# Improving Compositional Generalization using Iterated Learning and Simplicial Embeddings

**Yi Ren**
University of British Columbia*
renyi.joshua@gmail.com

**Samuel Lavoie**
Université de Montréal & Mila
samuel.lavoie.m@gmail.com

**Mikhail Galkin**
Intel AI Lab
mikhail.galkin@intel.com

**Danica J. Sutherland**
University of British Columbia & Amii
dsuth@cs.ubc.ca

**Aaron Courville**
Université de Montréal & Mila
aaron.courville@gmail.com

## Abstract

Compositional generalization, the ability of an agent to generalize to unseen combinations of latent factors, is easy for humans but hard for deep neural networks. A line of research in cognitive science has hypothesized a process, "iterated learning," to help explain how human language developed this ability; the theory rests on simultaneous pressures towards compressibility (when an ignorant agent learns from an informed one) and expressivity (when it uses the representation for downstream tasks). Inspired by this process, we propose to improve the compositional generalization of deep networks by using iterated learning on models with simplicial embeddings, which can approximately discretize representations. This approach is further motivated by an analysis of compositionality based on Kolmogorov complexity. We show that this combination of changes improves compositional generalization over other approaches, demonstrating these improvements both on vision tasks with well-understood latent factors and on real molecular graph prediction tasks where the latent structure is unknown.

## 1   Introduction

Deep neural networks have shown an amazing ability to generalize to new samples on domains where they have been extensively trained, approaching or surpassing human performance on tasks including image classification [62], Go [70], reading comprehension [13], and more. A growing body of literature, however, demonstrates that some tasks that can be easily solved by a human can be hard for deep models. One important such problem is compositional generalization ([18], *comp-gen* for short). For example, Schott et al. [65] study manually-created vision datasets where the true generating factors are known, and demonstrate that a wide variety of current representation learning methods struggle to learn the underlying mechanism. To achieve true "artificially intelligent" methods that can succeed at a variety of difficult tasks, it seems necessary to demonstrate compositional generalization. One contribution of this paper is to lay out a framework towards understanding and improving compositional generalization, and argue that most currently-common training methods fall short.

---

*Work done in part during an internship at Mila.

37th Conference on Neural Information Processing Systems (NeurIPS 2023).

In wondering how deep networks can learn to compositionally generalize, we might naturally ask: how did humans achieve such generalization? Or, as a particular case, how did human languages evolve components (typically, words) that can systematically combine to form new concepts? This has been a long-standing question in cognitive science and evolutionary linguistics. One promising hypothesis is known as *iterated learning* (IL), a procedure simulating cultural language evolution [41]. Aspects of this proposal are supported by lab experiments [42], a Bayesian model [7], the behavior of neural networks in a simple emergent communication task [60], and real tasks like machine translation [50] and visual question answering [76].

To link the study in cognitive science and deep learning, we first analyze the necessary properties of representations in order to generalize well compositionally. By linking the compositionality and the Kolmogorov complexity, we find iteratively resetting and relearning the representations can introduce compressibility pressure to the representations, which is also the key to the success of iterated learning. To apply iterated learning in a general representation learning problem, we propose to split the network into a backbone and a task head, and discretize the representation at the end of the backbone using *simplicial embeddings* (SEM, [45]). This scheme is more practical than LSTM [34] encoders previously used for neural iterated learning [60]. We observe in various controlled vision domains that SEM-IL can enhance compositional generalization by aligning learned representations to ground-truth generating factors. The proposed method also enhances downstream performance on molecular graph property prediction tasks, where the generating process is less clear-cut.

## 2 Compositional Generalization

Generalization is a long-standing topic in machine learning. The traditional notion of (in-distribution) generalization assumes that training and test samples come from the same distribution, but this is insufficient for many tasks: we expect a well-trained model to generalize to some novel scenarios that are unseen during training. One version of this is *compositional generalization* (comp-gen) [17], which requires the model to perform well on novel combinations of semantic concepts.

### 2.1 Data-generating assumption and problem definition

Any type of generalization requires some "shared rules" between training and test distributions. We hence assume a simple data-generating process that both training and test data samples obey. In Figure 1, the semantic generating factors, also known as latent variables, are divided into two groups: the task-relevant factors (or semantic generating factors) $\mathbf{G} = [G_1, ..., G_m]$, and task-irrelevant (or noise) factors $\mathbf{O}$. This division depends on our understanding of the task; for example, if we only want to predict the digit identity of an image in the color-MNIST dataset [3], then $m = 1$ and $G_1$ represents the digit identity. All the other generating factors such as color, stroke, angle, and possible noise are merged into $\mathbf{O}$. If we want to predict a function that depends on both identity and color, e.g. identifying blue even numbers, we could have $\mathbf{G} = [G_1, G_2]$ with $G_1$ the identity and $G_2$ the color.

Each input sample $\mathbf{x} \in \mathcal{X}$ is determined by a deterministic function $\mathsf{GenX}(\mathbf{G}, \mathbf{O})$. The task label(s) $\mathbf{y} \in \mathcal{Y}$ only depend on the factors $\mathbf{G}$ and possible independent noise $\epsilon$, according to the deterministic function $\mathsf{GenY}(\mathbf{G}, \epsilon)$. Note $(\mathbf{x}, \mathbf{O}) \perp\!\!\!\perp (\mathbf{y}, \epsilon) \mid \mathbf{G}$, and that $\mathbf{O}$, $\mathbf{G}$, and $\epsilon$ are independent. The data-generating distribution $P(\mathbf{x}, \mathbf{y})$ is determined by the latent distributions $P(\mathbf{G})$ and $P(\mathbf{O})$, along with the $\mathsf{GenX}$ and $\mathsf{GenY}$. We assume $\mathsf{GenX}$ and $\mathsf{GenY}$ are fixed across environments (the "rules of production" are consistent), while $P(\mathbf{G})$ and $P(\mathbf{O})$ might change between training and test.[2]

For compositional generalization, we wish to model the problem of generalizing to new combinations of previously seen attributes: understanding "red circle" based on having seen "red square" and "blue circle." Thus, we may assume that the supports of $P(\mathbf{G})$ are non-overlapping between train and test. (If this assumption is not true, it only makes the problem easier.) In summary, our goal is to find an algorithm $\mathcal{A}$ such that, when trained on a dataset $\mathcal{D}_{train} \sim P_{train}^n$, $\mathcal{A}$ achieves small test risk $\mathcal{R}_{P_{test}}(\mathcal{A}(\mathcal{D}_{train}))$. Here $P_{train}$ and $P_{test}$ should satisfy these conditions:

- $P_{train}$ and $P_{test}$ have $\mathbf{G}$, $\mathbf{O}$, $\epsilon$ jointly independent, and $\mathbf{x} = \mathsf{GenX}(\mathbf{G}, \mathbf{O})$, $\mathbf{y} = \mathsf{GenY}(\mathbf{G}, \epsilon)$.
- $\mathsf{GenX}$ and $\mathsf{GenY}$ are the same deterministic functions for $P_{train}$ and $P_{test}$.
- In challenging cases, we may have $\mathsf{supp}[P_{train}(\mathbf{G})] \cap \mathsf{supp}[P_{test}(\mathbf{G})] = \emptyset$.

---

[2]This differs from the classical setting of covariate shift: $P(\mathbf{y} \mid \mathbf{x})$ might change due to the shift in $P(\mathbf{G})$.

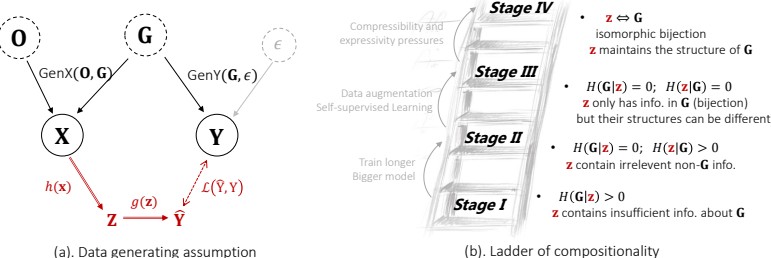

(a). Data generating assumption    (b). Ladder of compositionality

Figure 1: Left: the data-generating assumption and a typical representation learning method (in red). We use the model $(g \circ h)(\mathbf{x})$ for downstream predictions. Right: the ladder of compositionality stating the requirements of $\mathbf{z}$ using the entropy-related measurements; see Appendix A for more.

## 2.2 Representationl Learning and Ladder of Compositionality

For compositional generalization, we expect that the model must extract atomic semantic features from the training data, and systematically re-combine them in a procedure akin to how the data is generated [41]. We thus consider a typical representation learning framework, which resembles the inverse of the data generation process (Figure 1(a), bottom). We use a backbone $h : \mathcal{X} \to \mathcal{Z}$ to convert the input signal $\mathbf{x}$ into a representation $\mathbf{z}$, and a task head $g : \mathcal{Z} \to \mathcal{Y}$ to solve the given task based on that representation $\mathbf{z}$. The prediction of the model is $\hat{\mathbf{y}} = (g \circ h)(\mathbf{x})$.

Intuitively, we would like our learned $\mathbf{z}$ to uncover the hidden $\mathbf{G}$, and $g(\mathbf{z})$ to recover $\mathsf{GenY}(\mathbf{G}, \epsilon)$. We thus analyze how the relationship between $\mathbf{z}$ and $\mathbf{G}$ influences the model's generalization capability, building off principles such as information bottleneck [74]. Inspired by the "ladder of causation" [55], we propose a "ladder of compositionality" in Figure 1(b), which outlining a series of conditions on $\mathbf{z}$ and $\mathbf{G}$. We hypothesize that comp-gen roughly requires reaching the highest rung of that ladder:

**Hypothesis 1.** *To generalize compositionally, the learned $\mathbf{z}$ should capture exactly the information in $\mathbf{G}$ and nothing more ($\mathbf{G}$ to $\mathbf{z}$ should be a bijection), and moreover it should preserve the "structure" of $\mathbf{G}$ (i.e. the mapping from $\mathbf{G}$ to $\mathbf{z}$ should be an isomorphism).*

More on this hypothesis, the ladder, and relationship to models of disentanglement [32] are discussed in Appendix A. In short, we find that a model trained using common learning methods relying on mutual information between input $\mathbf{x}$ and supervision $\mathbf{y}$ cannot reliably reach the final stage of the ladder – it is necessary to seek other inductive biases in order to generalize compositionally.

## 3 Compressibility pressure and Compositional mapping

From the analysis above, we need to find other inductive biases to obtain compositional mappings. Inspired by how compositionality emerges in human language,[3] we speculate that the *compressibility* pressure is the key. Note that this pressure does not refer to compressing information from $\mathbf{x}$ to $\mathbf{z}$ (as in Stage III does), but whether a mapping can be expressed in a compact way by reusing common rules. In this section, we will first link compressibility pressure to Kolmogorov complexity by defining different mappings using group theory. As the Kolmogorov complexity is hard to compute, making explicit regularization dificult, we propose to implicitly regularize via *iterated learning*, a procedure in cognitive science proposed to increase compositionality in human-like language.

### 3.1 Compositional mappings have lower Kolmogorov complexity

From Occam's razor, we know efficient and effective mappings are more likely to capture the ground truth generating mechanism of the data, and hence generalize better. The efficiency is determined by how compressed the mapping is, which can also be measured by Kolmogorov complexity [47, 71].

---

[3]Human languages are examples of compositional mapping [35]: words are composed of combinations of reusable morphemes, and those words in turn are combined to form complex sentences following specific *stable rules*. These properties make our language unique among natural communication systems and enable humans to convey an open-ended set of messages in a compositional way [42]. Researchers in cognitive science and evolutionary linguistics have proposed many explanations for the origin of this property; one persuasive method for simulating it is iterated learning [41].

To build a link between compositionality and Kolmogorov complexity, we can first describe different bijections between **z** and **G** using group theory, and then use the description length to compare the complexity of a typical element. Specifically, assuming $\mathbf{z} \in \mathcal{Z}$, $\mathbf{G} \in \mathcal{G}$ and $|\mathcal{Z}| = |\mathcal{G}|$, the space of all bijections between **z** and **G** is an isomorphism of a symmetric group $S_{|\mathcal{G}|}$. If $\mathbf{G} = [G_1, ..., G_m]$ and each $G_m$ has $v$ different possible values, $|\mathcal{G}| = v^m$. For clarity in the analysis, we assume **z** also has the same shape. Then, any bijection between **z** and **G** can be represented by an element in $S_{v^m}$.

The space of compositional mapping, which is a subset of all bijections, has more constraints. Recall how a compositional mapping is generated (see Appendix A.4 for more details): we first select $z_i$ for each $G_j$ in a non-overlapping way. Such a process can be represented by an element in $S_m$. After that, we will assign different "words" for each $z_i$, which can be represented by an element in $S_v$. As we have $m$ different $z_i$, this procedure will be repeated $m$ times. In summary, any compositional mapping can be represented by an element in the group $S_v^m \rtimes S_v \in S_{v^m}$, where $\rtimes$ is the semidirect product in group theory. The cardinality of $S_{v^m}$ is significantly larger than $S_v^m \rtimes S_v$, and so a randomly selected bijection is unlikely to be compositional. Thus

**Proposition 1** (Informal). *For $m, v \geq 2$, among all bijections, any compositional mapping has much lower Kolmogorov complexity than a typical non-compositional mapping.*

We prove this by constructing descriptive protocols for each bijection. As a compositional mapping has more *reused rules*, its description length can be smaller (see Appendix B.1 for more details).

### 3.2 Compressibility pressure is amplified in iterated learning

Now, our target is finding bijections with higher compositionality and lower Kolmogorov complexity, which are both non-trivial. Because the ground truth **G** is usually inaccessible and the Kolmogorov complexity is hard to calculate. Fortunately, researchers find that human language also evolved to become more compositional without knowing **G**. Authors of [42] hypothesize that the *compressibility pressure*, which exists when an innocent agent (e.g., a child) learns from an informed agent (e.g., an adult), plays an important role. Such pressure is reinforced and amplified when the human community repeats this learning fashion for multiple generations.

However, the aforementioned hypothesis assumes that simplicity bias is inborn in the human cognition system. Will deep neural agents also have similar preferences during training? The answer is yes. By analyzing an overparameterized model on a simple supervised learning problem, we can strictly prove that repeatedly introducing new agents to learn from the old agent (then this informed agent becomes the old agent for the next generation) can exert a non-trivial regularizing effect on the number of "active bases" of the learned mapping. Restricting the number of active bases encourages the model to reuse the learned rules. In other words, this regularization effect favors mappings with lower Kolmogorov complexity, which is exactly what we expect for compositional generalization. Due to the space limits, we left the formulation and proof of this problem in Appendix B.2.

### 3.3 Complete the proposed solution

We thus expect that iteratively resetting and relearning can amplify the compressibility pressure, which helps us to reach the final rung of the ladder from the third. Before that, we need another pressure to reach third rung (i.e., ensure a bijection between **z** and **G**). *Expressivity pressure*, constraining the learned mapping to be capable enough to accomplish the downstream tasks, is what we need.

The complete iterated learning hypothesis of Kirby et al. [42] claims that the compositional mapping emerges under the interaction between the compressibility pressure (i.e., efficiency) and the expressivity pressure (i.e., effectiveness). Inspired by this, we propose to train a model in generations consisting of two phases. At the $t$-th generation, we first train the backbone $h$ in an *imitation phase*, where a student $h_t^S$ learns to imitate **z** sampled from a teacher $h_t^T$. As analyzed above, iteratively doing so will amplify the compressibility pressure. Then, in the following *interaction phase*, the model $g_t \circ h_t^S$ follows standard downstream training to predict **y**. The task head $g_t$ is randomly initialized and fine-tuned together with the backbone in this phase. By accomplishing this phase, the expressivity pressure is introduced. The fine-tuned backbone $h_t^S$ then becomes the teacher $h_{t+1}^T$ for the next generation, and we repeat, as illustrated in Figure 2 and Algorithm 1.

Another problem with applying iterated learning to deep neural networks is how to create the discrete message, i.e., **z**. Discretization is not *necessary*: for example, the imitation phase could use $L_2$ loss to

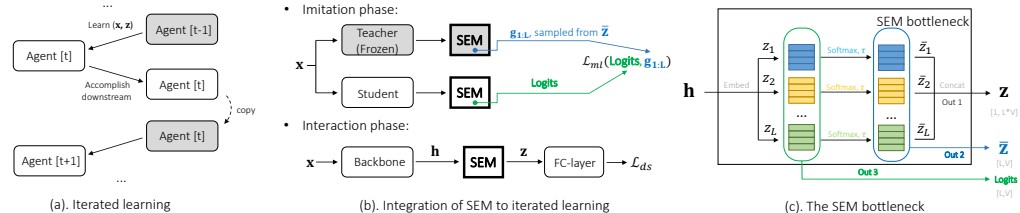

Figure 2: An illustration of iterated learning and SEM layer design.

match a student's continuous representations to the teacher's. We find greatly improved performance with our discretization scheme, however, due to much-increased compressibility pressure. It is also possible [60] to use e.g. an LSTM encoder at the end of $h(\mathbf{x})$ to produce discrete $\mathbf{z}$, and an LSTM decoder at the start of $g(\mathbf{z})$. The interaction phase is then not directly differentiable; though many estimator options exist [6, 39, 78], training tends to be difficult due to high bias and/or variance.

Instead, we consider a simplicial embedding layer (SEM, [45]), which has proven effective on many self-supervised learning tasks. As illustrated in Figure 2(c), a dense representation $\mathbf{h}$ (the output of the original backbone) is linearly transformed into $m$ vectors $z_i \in \mathbb{R}^v$. Then we apply a separate softmax with temperature $\tau$ to each $z_i$, yielding $\bar{z}_i$ which are, if the temperature is not too high, approximately sparse; the $\bar{z}_i$ are then concatenated to a long vector $\mathbf{z}$. The overall process is

$$\bar{z}_i = \mathsf{Softmax}_\tau(z_i) = \left[ \frac{e^{z_{ij}/\tau}}{\sum_{k=1}^V e^{z_{ik}/\tau}} \right]_j \in \mathbb{R}^v \qquad \mathbf{z} = \begin{bmatrix} \bar{z}_1^\top & \dots & \bar{z}_m^\top \end{bmatrix}^\top \in \mathbb{R}^{mv}. \qquad (1)$$

By using an encoder with a final SEM layer, we obtain an approximately-sparse $\mathbf{z}$. In the imitation phase, we generate discrete pseudo-labels by sampling from the categorical distribution defined by each $\bar{z}_i$, then use cross-entropy loss so that the student is effectively doing multi-label classification to reconstruct the teacher's representations. In the imitation phase, the task head $g$ operates directly on the long vector $\mathbf{z}$. The full model $g \circ h$ is differentiable, so we can use any standard task loss. Pseudocode for the proposed method, SEM-IL, is in the appendix (Algorithm 1).

# 4    Analysis on Controlled Vision Datasets

We will first verify the effectiveness of the proposed SEM-IL method on controlled vision datasets, where the ground truth $\mathbf{G}$ is accessible. Thus, we can directly observe how $\mathbf{z}$ gradually becomes more similar to $\mathbf{G}$, and how the compressibility and expressivity pressures affect the training process. In this section, we consider a regression task on 3dShapes [9], where recovering and recombining the generating factors is necessary for systematic generalization. The detailed experimental settings and results on additional similar datasets, dSprites [52] and MPI3D-real [23], are given in Appendix C.

## 4.1    The Effectiveness of SEM-IL

**Better comp-gen performance**    We first show the effectiveness of the proposed method using results on 3dShapes, containing images of objects with various colors, sizes, and orientations against various backgrounds. Here $\mathbf{G}$ numerically encodes floor hue, wall hue, object hue, and object scale into discrete values, and the goal is to recover a particular linear function of that $\mathbf{G}$. (Results for a simple nonlinear function were comparable.)

We compare five algorithms:

- Baseline: directly train a ResNet18 [31] on the downstream task.
- SEM-only: insert an SEM layer to the baseline model.
- IL-only: train a baseline model with Algorithm 1, using MSE loss during imitation.
- SEM-IL: train an SEM model with Algorithm 1.
- Given-G: train an SEM model to reproduce the true $\mathbf{G}$ (which would not be known in practice), then fine-tune on the downstream task.

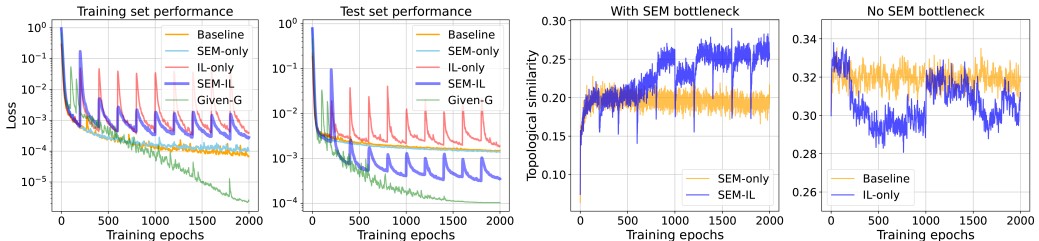

Figure 3: Left: compositional generalization performance on a regression task. Right: topological similarity for IL and non-IL methods. Note the values of $\rho$ in the two panels are not comparable, as the structure of $\mathbf{z}$ in the two settings (with or without SEM) is different.

In the first panel of Figure 3, we see that the baseline and SEM-only models perform similarly on the training set; IL-based methods periodically increase in error at the beginning of each generation, but are eventually only slightly worse than the baselines on training data. On the test set, however, evaluating compositional generalization by using values of $\mathbf{G}$ which did not appear in training, SEM-IL brings significant improvement compared with other methods. Using only SEM or only IL gives no improvement over the baseline, however; it is only their combination which helps, as we will discuss further shortly. The (unrealistic) oracle method Given-G is unsurprisingly the best, since having $\mathbf{z}$ similar to $\mathbf{G}$ is indeed helpful for this task.

**How z evolves during learning** To see if better generalization ability is indeed achieved by finding $\mathbf{z}$ that resembles the structure of $\mathbf{G}$, we check their topological similarity[4]

$$\rho(\mathbf{z}, \mathbf{G}) \triangleq \text{Corr}\left( d_z(\mathbf{z}^{(i)}, \mathbf{z}^{(j)}), d_G(\mathbf{G}^{(i)}, \mathbf{G}^{(j)}) \right) \tag{2}$$

where $d_z$ and $d_G$ are distance metrics, $\mathbf{z}^{(i)}$ is the predicted representation of $\mathbf{x}^{(i)}$, and $\mathbf{G}^{(i)}$ is the corresponding ground-truth generating factors. This measurement is widely applied to evaluate the compositionality of the mappings in cognitive science [8] and emergent communication [60]. Following existing works, we use the Hamming distance for $\mathbf{G}$ and discretized $\mathbf{z}$ in SEM-based methods, and cosine distance for continuous $\mathbf{z}$ in non-SEM methods. We expect $h(\mathbf{x})$ to map $\mathbf{x}$ with similar $\mathbf{G}$ to close $\mathbf{z}$, and dissimilar $\mathbf{x}$ to distant $\mathbf{z}$, so that $\rho(\mathbf{z}, \mathbf{G})$ will be high.

The third panel of Figure 3 shows that the SEM-only model quickly reaches a plateau after 200 epochs and then slowly decreases, while SEM-IL, after briefly stalling at the same point, continues to increase to a notably higher topological similarity. In the last panel, however, the IL-only method doesn't improve $\rho$ over the baseline: it seems both parts are needed.

## 4.2 Discretized Representation is Beneficial for the Imitation Phase of IL

To explain why SEM and IL cooperate well, we need to look deeper into how the compressibility pressure influences the learning of representations. This pressure induced by iterated learning, which helps us to find mappings with lower Kolmogorov complexity, leads to representations that are more compositional and systematic [42]. However, in prior works, these mappings were only considered in conjunction with some discretized representation [54, 60]. While IL could be used with continuous representation during the imitation phase, similar to born-again networks [19], we found that our algorithm benefits a lot from the discretized representations.

To get a clear picture of why discretized representations are so important, we divide $h(\mathbf{x})$ into $m$ sub-mappings $h_i(\mathbf{x})$, which map $\mathbf{x}$ to $\bar{z}_i \in [0, 1]^v$. We can understand each $\bar{z}_i$ as a categorical distribution over $v$ different possible values. As such, during training, the model learns discrete features of the dataset and assigns confidence about each feature for every sample. The neural network will tend to more quickly learn simpler mappings [5, 24], and will assign higher confidence according to the mapping it has learned. In other words, if a mapping does not align well with $\mathbf{G}$, it is more likely to give idiosyncratic learned $\bar{z}_i$, and will lead to low confidence for most samples. On the contrary, $\bar{z}_i$ belonging to compositional mappings will be more general, and on average tend towards higher confidence.

---

[4]This measure is also known as the distance correlation [72]; it is a special case of the Hilbert-Schmidt Independence Critierion (HSIC, [25]) for a particular choice of kernel based on $d_z$ and $d_G$ [66].

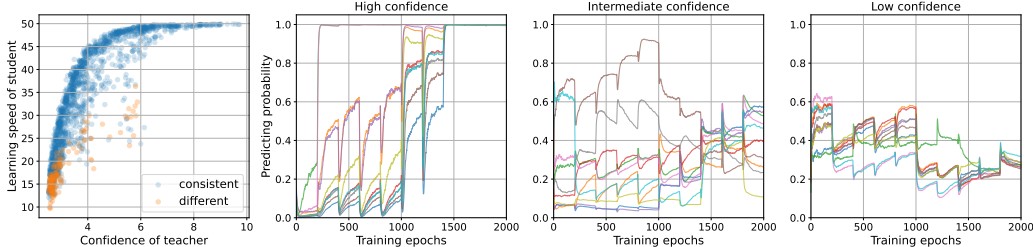

Figure 4: First panel: correlation between teacher's confidence and student's learning speed for each $(\mathbf{x}, \bar{z}_i)$; $\bar{z}_i$ is the prediction of the $l$-th attribute in imitation phase. "Consistent" means the student makes the same prediction as the teacher. Other panels: learning curves of the student's predictions.

The imitation phase reinforces this bias when the new student learns from the *sampled* pseudo labels $\mathbf{g}_i$ from the teacher's prediction $\bar{z}_i$. As such, confident predictions, which are more likely to belong to the compositional mappings, will be learned faster (and harder to forget) by the student. On the contrary, for less confident features where $P(\bar{z}_i \mid \mathbf{x})$ is flat, $\mathbf{g}_i$ could change across epochs. This makes it hard for the student to remember any related $(\mathbf{x}, \mathbf{g}_i)$. For example, a student will be reluctant to build a stable mapping between "red" and $z_1$ if the teacher communicates ("red square", $\mathbf{g}_1 = 0$), ("red square", $\mathbf{g}_1 = 1$), ("red square", $\mathbf{g}_1 = 2$) in three consecutive epochs.

Furthermore, using the sampled pseudo-labels can help the student to align the learned $(\mathbf{x}, \mathbf{g}_i)$ better. Assume during training, the student already remembers some pairs like ("blue circle", $\mathbf{g}_1 = 0$), ("blue square", $\mathbf{g}_1 = 0$), ("blue star", $\mathbf{g}_1 = 0$), but the teacher is not confident in ("blue apple", $\mathbf{g}_1$), perhaps because apples are rarely blue. Following the analysis above, as $P(\bar{z}_1 \mid \text{blue apple})$ is flat, the teacher may generate $\mathbf{g}_1 \neq 0$ a significant portion of the time. However, if the teacher happens to generate $\mathbf{g}_1 = 0$ at some point, the student would learn ("blue apple", $\mathbf{g}_1 = 0$) faster than those with $\mathbf{g}_1 \neq 0$, because it aligns well with the other information stored in the student network. The parameter updates caused by the learning of other ("blue [shape]", $\mathbf{g}_1 = 0$) will also promote the learning of ("blue apple", $\mathbf{g}_1 = 0$), similar to how "noisy" labels are fixed as described by [61].

To support the explanations above, we can first observe the correlation between the teacher's confidence and the model's learning speed for $(\mathbf{x}, \mathbf{g}_i)$. Specifically, for each $\mathbf{x}$, the teacher makes $m$ predictions with the corresponding categorical distribution $\bar{z}_i$, $i \in [m]$. For each $(\mathbf{x}, \bar{z}_i)$, the confidence is measured by the negative logarithm of the teacher's predicted probability, $-\log[\bar{z}_i]_{\hat{j}}$ where $\hat{j} \in \text{argmax}_j[\bar{z}_i]_j$. The learning speed of $(\mathbf{x}, \mathbf{g}_i)$ is measured by the integral of the student's prediction with training time $t$, i.e., $\sum_{t=0}[\hat{z}_i(t)]_j$, where $j$ is the value provided by the teacher and $\hat{z}_i(t)$ is student's prediction at time $t$. As illustrated in the first panel of Figure 4, the $(\mathbf{x}, \mathbf{g}_i)$ with higher confidence are usually learned faster by the student.

We also provide the learning curves of $(\mathbf{x}, \mathbf{g}_i)$ with high/intermediate/low confidence (each with 10 samples) in the other three panels of the figure. The curves for high-confidence samples all converge to $[\hat{z}_i(t)]_j = 1$ while those for low-confidence predictions could converge to a value less than 0.3. This means the student might make predictions that are different from the teacher's supervision. By highlighting such low-confidence $(\mathbf{x}, \mathbf{g}_i)$ pairs in the scatter plot, we find they are all low-confidence samples. Another interesting observation from the high-confidence curves is that some $(\mathbf{x}, \mathbf{g}_i)$ pairs are not remembered by the student in the first generation: they emerge at some point and gradually dominate as the training goes on. This phenomenon matches our analysis of how the sampled pseudo-labels help the student align $(\mathbf{x}, \mathbf{g}_i)$ to its knowledge well. To further support this explanation, Appendix C shows that performance is substantially harmed by taking pseudo-labels from the argmax, rather than sampling from $\bar{z}_i$.

To recap, this subsection provided an explanation (along with some supporting evidence) for why the combination of SEM and IL is so important, based on the perspective of sample difficulty, which we believe to be a significant factor in the success of this algorithm.

## 5 Application: Molecular Property Prediction

Given the success in controlled vision examples, we now turn to a real problem where the true generative process is unknown. We focus on predicting the properties of molecular graphs, for several

reasons. First, molecular graphs and their labels might follow a (chemical) procedure akin to that in Figure 1: for instance, one $G_i$ might be the existence of a specific functional group, or the number of specific atoms. Different molecular properties could then be determined by different subsets of $G_i$, as we desired in the compositional generalization problem. Furthermore, the generating mechanisms (GenX and GenY) should be consistent and determined by nature. Second, benchmark datasets in this community contain various types of tasks (e.g., binary classification, multi-label classification, and regression) with similar input signals: performing well on different tasks will broaden the scope of our algorithm. Furthermore, the scaffold split used by most molecular datasets corresponds well to the compositional generalization setup we consider here. (We also try some more challenging splits, using structural information.) Last, learning meaningful representations that uncover the generating mechanisms of molecules is important, of practical significance, and difficult: it can potentially help predict the properties of unknown compounds, or accelerate the discovery of new compounds with specific properties, but scaling based on massive datasets as in recent work on vision or language seems more difficult. We hope our analysis can provide a new perspective on this problem.

## 5.1 Improvement on the Downstream Performance

Table 1: Downstream performance on different tasks. The numbers of AUROC and average precision are in percent form. For PCQM, we report the validation performance, as the test set is private and inaccessible. Means and standard deviations of 5 seeds are given. Valid/test-full means the standard train/val/test split provided by the dataset. Valid/test-half means we train the model on half of the training data which is *less similar* to the validation and test sets. See Appendix D for more.

| Model and Algorithm | | molhiv (AUROC ↑) | | | | molpcba (Avg.Precision ↑) | | | | PCQM (MAE ↓) |
|---|---|---|---|---|---|---|---|---|---|---|
| | | Valid-full | Test-full | Valid-half | Test-half | Valid-full | Test-full | Valid-half | Test-half | Valid |
| GCN | Baseline | 82.41±1.14 | 76.25±0.38 | 75.65±0.91 | 72.31±1.86 | 21.44±0.25 | 22.13±0.46 | 21.13±0.38 | 20.78±0.62 | 0.125±0.002 |
| | Baseline+ | 81.61±0.63 | 75.58±1.00 | 73.23±0.75 | 72.17±1.02 | 22.31±0.34 | 22.68±0.30 | 21.01±0.45 | 20.60±0.37 | 0.118±0.004 |
| | SEM-only | 84.00±1.10 | 78.40±0.67 | 74.84±1.57 | 72.81±2.32 | 26.39±0.66 | 25.89±0.71 | **22.79±0.91** | **22.09±1.02** | 0.106±0.002 |
| | **SEM-IL** | **84.89±0.68** | **79.09±0.67** | **78.48±0.67** | **74.02±0.78** | **28.81±0.72** | **27.15±0.74** | 22.59±0.84 | 21.90±0.81 | **0.102±0.005** |
| GIN | Baseline | 81.76±1.04 | 76.99±1.42 | 76.95±1.40 | 71.63±2.21 | 23.09±0.32 | 22.64±0.49 | 20.52±0.39 | 20.15±0.42 | 0.109±0.003 |
| | Baseline+ | 81.55±0.72 | 77.01±0.94 | 74.77±1.62 | 69.75±3.10 | 23.85±0.29 | 22.91±0.40 | 21.71±0.12 | 20.98±0.27 | 0.108±0.003 |
| | SEM-only | 83.05±0.90 | 78.21±0.78 | 76.29±2.06 | 72.70±4.94 | 26.01±0.52 | 25.66±0.47 | 22.26±0.39 | 21.50±0.48 | 0.106±0.004 |
| | **SEM-IL** | **83.32±1.51** | **78.61±0.73** | **78.06±1.24** | **72.89±0.48** | **29.30±0.48** | **28.02±0.61** | **24.41±0.47** | **23.89±0.77** | **0.098±0.005** |

We conduct experiments on three common molecular graph property datasets: ogbg-molhiv (1 binary classification task), ogbg-molpcba (128 binary classification tasks), and PCQM4Mv2 (1 regression task); all three come from the Open Graph Benchmark [37]. We choose two types of backbones, standard GCN [40] and GIN [80]. For the baseline experiments, we use the default hyperparameters from [37]. As the linear transform added in SEM-based method gives the model more parameters, we consider "baseline+" to make a fair comparison: this model has an additional embedding layer, but no softmax operation. Detailed information on these datasets, backbone models, and hyper-parameters is provided in Appendix D.

From Table 1, we see the SEM-IL method almost always gives the best performance. Unlike in the controlled vision experiments (Figure 3), however, SEM alone can bring significant improvements in this setting. We speculate that compressibility pressure might be more significant in the interaction phase (i.e. standard training) when the generating mechanism is complex. This suggests it may be possible to develop a more efficient algorithm to better impose compressibility and expressivity pressures at the same time.

## 5.2 Probing Learned z by Meaningful Structures

In the controlled vision examples, we know that SEM-IL not only enhances the downstream performance, but also provides **z** more similar to the ground-truth generating factors, as seen by the improvement in topological similarity. However, as the generating mechanism is usually inaccessible in real problems, we indirectly measure the quality of **z** using graph probing [2]. Specifically, we first extract some meaningful substructures in a molecule using domain knowledge. For example, we can conclude whether a benzene ring exists in **x** by directly observing its 2D structure. With the help of the RDKit tool [44], we can generate a sequence of labels for each **x**, which is usually known as the "fingerprint" of molecules (denoted $FP(\mathbf{x}) \in \{0,1\}^k$, indicating whether each specific structure exists in **x**). Then, we add a linear head on top of the fixed **z** and train it using a generated training set $(\mathbf{x}, FP(\mathbf{x})), \mathbf{x} \sim \mathcal{D}_{train}$, and compare the generalization performance on the generated test set

$(\mathbf{x}, \mathsf{FP}(\mathbf{x})), \mathbf{x} \sim \mathcal{D}_{test}$. For fair comparison, we set $m = 30$ and $v = 10$ to make $\mathbf{z}$ and $\mathbf{h}$ be the same width, excluding the influence of the linear head's capacity.

Table 2: AUROC for graph probing based on different $\mathbf{z}$; random guessing would be $\approx 0.5$.

| | | Sat.Ring | Aro.Ring | Aro.Cycle | Aniline | Ketone | Bicyc. | Methoxy | ParaHydrox. | Pyridine | Benzene | Avg. |
|---|---|---|---|---|---|---|---|---|---|---|---|---|
| | Init. base | 0.870 | 0.958 | 0.811 | 0.629 | 0.595 | 0.615 | 0.627 | 0.706 | 0.692 | 0.812 | 0.732 |
| | Init. SEM | 0.872 | 0.958 | 0.812 | 0.635 | 0.597 | 0.638 | 0.613 | 0.692 | 0.683 | 0.815 | 0.731 |
| Train on Molhiv | Baseline | 0.874 | 0.948 | 0.916 | 0.700 | 0.717 | 0.694 | 0.804 | 0.740 | 0.703 | 0.913 | 0.801 |
| | SEM-only | 0.893 | **0.989** | 0.938 | 0.722 | 0.751 | 0.779 | 0.823 | 0.763 | 0.763 | 0.938 | 0.836 |
| | **SEM-IL** | **0.907** | 0.980 | **0.967** | **0.781** | **0.801** | **0.794** | **0.903** | **0.815** | **0.869** | **0.965** | **0.878** |
| Train on Molpcba | Baseline | 0.921 | 0.988 | 0.968 | 0.866 | 0.875 | 0.835 | 0.875 | 0.855 | 0.856 | 0.968 | 0.901 |
| | SEM-only | **0.942** | **0.991** | 0.981 | 0.888 | 0.916 | **0.854** | **0.921** | 0.888 | 0.897 | 0.980 | 0.926 |
| | SEM-IL | 0.940 | 0.988 | **0.982** | **0.910** | **0.931** | 0.849 | 0.912 | **0.910** | **0.912** | **0.981** | **0.931** |
| Train on 10% pcba | Baseline | 0.923 | 0.980 | 0.962 | 0.863 | 0.857 | 0.832 | 0.870 | 0.833 | 0.864 | 0.962 | 0.895 |
| | SEM-only | **0.943** | 0.993 | **0.989** | 0.872 | 0.906 | 0.835 | 0.913 | **0.876** | 0.900 | **0.989** | 0.922 |
| | SEM-IL | 0.938 | **0.994** | 0.985 | **0.891** | **0.918** | **0.847** | **0.927** | 0.874 | **0.907** | 0.985 | **0.927** |
| Train on pcba-1task | Baseline | 0.892 | 0.974 | 0.948 | 0.723 | 0.750 | 0.689 | 0.845 | 0.758 | 0.782 | 0.947 | 0.831 |
| | SEM-only | 0.906 | **0.989** | 0.958 | **0.772** | 0.809 | 0.735 | 0.876 | **0.770** | 0.835 | 0.957 | 0.861 |
| | SEM-IL | **0.906** | 0.988 | **0.963** | 0.741 | **0.851** | **0.744** | **0.887** | 0.765 | **0.869** | **0.962** | **0.867** |

In the experiments, we use the validation split of molhiv as $\mathcal{D}_{train}$ and the test split as $\mathcal{D}_{test}$, each of which contain 4,113 distinct molecules unseen during the training of $\mathbf{z}$. The generalization performance of ten different substructures is reported in Table 2. The first block (first two rows) of the table demonstrates the performance of two types of models before training. They behave similarly across all tasks and give a higher AUROC than a random guess. Then, comparing the three algorithms in each block, we see SEM-based methods consistently outperform the baseline, which supports our hypothesis well. SEM-IL outperforms SEM-only on average, but not for every task; this may be because some structures are more important to the downstream task than others.

Comparing the results across the four blocks, we find that the task in the interaction phase also influences the quality of $\mathbf{z}$: the $\mathbf{z}$ trained by molpcba is much better than those trained by molhiv. To figure out where this improvement comes from, we first use only 10% of the training samples in molpcba to make the training sizes similar, then make the supervisory signal more similar by using only one task from molpcba. As illustrated in the last two blocks in the table, we can conclude that the complexity of the task in the interaction phase, which introduces the expressivity pressure, plays a more important role in finding better $\mathbf{z}$.

Based on this observation, we can improve SEM-IL by applying more complex interaction tasks. For example, existing works on iterated learning use a referential game or a reconstruction task in the interaction phase, which could introduce stronger expressivity pressure from a different perspective. Furthermore, [45] demonstrates that SEM works well with most contrastive learning tasks. We hope the fundamental analysis provided in this paper can shed light on why SEM and IL collaborate so well and also arouse more efficient and effective algorithms in the future.

## 6 Related Works

**Iterated Learning and its Applications.** Iterated learning (IL) is a procedure that simulates cultural language evolution to explain how the compositionality of human language emerges [41]. In IL, the knowledge (i.e., the mapping between the input sample and its representation) is transferred between different generations, during which the compositional mappings gradually emerge and dominate under the interaction between compressibility and expressivity pressures. Inspired by this principle, there are some successful applications in symbolic games [60], visual question answering [76], machine translation [50], multi-label learning [58], reinforcement learning [54], etc.

There are also many algorithms training a neural network for multiple generations, which could possibly support the principles proposed in iterated learning. For example, [19] proposes to iteratively distill the downstream logits from the model in the previous generation, and finally bootstrap all the models to achieve better performance on image classification task; this can be considered as an IL algorithm merging the imitation and interaction phases together. [82] proposes to re-initialize the latter layers of a network and re-train the model for multiple generations, which is similar to an IL algorithm that only re-initializes the task head. [54] extends such a reset-and-relearn training to reinforcement learning and shows that resetting brings benefits that cannot be achieved by other regularization methods such as dropout or weight decay. In the era of large language models, self-refinement in-context learning [51] and self-training-based reinforcement learning [26] can also

benefit from iteratively learning from the signals generated by agents in the previous generation. We left the discussion and analysis on these more complex real systems in our future work.

**Knowledge Distillation and Discrete Bottleneck.** Broadly speaking, the imitation phase in SEM-IL, which requires the student network to learn from the teacher, can be considered as a knowledge distillation method [33]. Different from the usual setting, where the student learns from the teacher's prediction on a downstream task, we assume a data-generating mechanism and create a simplex space for the generating factors. By learning from the teacher in this space, we believe the compressibility pressure is stronger and is more beneficial for the compositional generalization ability.

For the discretization, there are also other possible approaches, e.g., [28] uses an LSTM to create a discrete message space, and [48] proposes a method using a vector quantized bottleneck [75]. We choose SEM [45] for its simplicity and universality: it is easy to insert it into a model for different tasks. Besides, SEM has proved to be effective on self-supervised learning tasks; we extend it to classification, regression, and multi-label tasks.

**Compressibility, learning dynamics, and Kolmogorov complexity** Recently, with the success of large language models, the relationship between compressibility and generalization ability gradually attracted more attention [12]. Authors of [57] propose that how well a model is compressed corresponds to the integral of the training loss curve when negative logarithmic likelihood loss is used. Although this claim assumes the model sees each training sample only once, which might not be consistent with the multiple-epochs training discussed in this paper, the principles behind this claim and our analysis are quite consistent: the mappings generalize better and are usually learned faster by the model. Furthermore, authors of [71] link the generalization ability to Kolmogorov complexity. Our analysis in Appendix B also supports this claim well. Hence we believe the evolution of the human cognition system can provide valuable insights into deep learning systems.

**Graph Representation Learning.** Chemistry and molecular modeling are some of the main drivers of neural graph representation learning since its emergence [21] and graph neural networks, in particular. The first theoretical and practical advancements [27, 40, 80] in the GNN literature were mostly motivated by molecular use cases. Furthermore, many standard graph benchmarks [15, 16, 37] include molecular tasks on node, edge, and graph-levels, e.g., graph regression in ZINC and PCQM4Mv2 or molecular property prediction in ogbg-molhiv and ogbg-molpcba datasets. Graph Transformers [14, 43, 59] exhibit significant gains over GNNs in molecular prediction tasks. Self-supervised learning (SSL) on graphs is particularly prominent in the molecular domain highlighted by the works of GNN PreTrain [38], BGRL [73], and Noisy Nodes [22]. We will extend the proposed method to different models and different pretraining strategies in our future work.

## 7 Conclusion

In this paper, we first define the compositional generalization problem by assuming the samples in the training and test sets share the same generating mechanism while the generating factors of these two sets can have different distributions. Then, by proposing the compositionality ladder, we analyze the desired properties of the representations. By linking the compositionality, compressibility, and Kolmogorov complexity together, we find iterated learning, which is well-studied in cognitive science, is beneficial for our problem. To appropriately apply iterated learning, we attach an SEM layer to the backbone model to discretize the representations. On the datasets where the true generating factors are accessible, we show that the representations learned by SEM-IL can better portray the generation factors and hence lead to better test performance. We then extend the proposed algorithm to molecular property prediction tasks and find it improves the generalization ability.

The main drawback of the current solution is the time-consuming training: we must run multiple generations and some common features might be re-learned multiple times, which is inefficient. Hence a more efficient way of imposing compressibility is desired.

Overall, though, our analysis and experiments show the potential of the SEM-IL framework on compositional generalization problems. We believe a better understanding of where the compressibility bias comes from in the context of deep learning can inspire more efficient and non-trivial IL framework designs. Clearly defining the compositional generalization problem and finding more related practical applications can also promote the development of IL-related algorithms.

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

**Algorithm 1:** Proposed IL-SEM algorithm

---

Split the network into $h(\mathbf{x})$ and $g(\mathbf{z})$, then add an SEM bottleneck to discretize $\mathbf{z}$
**for** $t = 0$ **to** $T_{gen}$ **do**
    Initialize the student speaker $h_t^S(\mathbf{x})$[5]
    *#       Imitation Phase (start from the second generation)*
    **if** $t > 0$ **then**
        **for** $i = 0$ **to** $I_{imit}$ **do**
            Sample a batch $\mathbf{x}$ from the training set $\mathcal{D}_{train}$
            Sample the pseudo labels from teacher's prediction (one-hot vectors) $\mathbf{g}_{1:m} \sim h_t^T(\mathbf{x})$
            Calculate the student's prediction $\mathbf{z} = [\hat{z}_1, ..., \hat{z}_m] = h_t^S(\mathbf{x})$
            Update $h_t^S(\cdot)$ with multi-label cross-entropy loss $\mathcal{L}_{ml} = \sum_{i=1}^m \mathbf{g}_i^\top \cdot \log \hat{z}_i$
        **end for**
    **end if**
    *#       Interaction Phase (regular training on the downstream task)*
    **for** $i = 0$ **to** $I_{int}$ **do**
        Initialize the listener $g_t(\mathbf{z})$ randomly
        Sample a batch $(\mathbf{x}, \mathbf{y})$ from the training set $\mathcal{D}_{train}$
        Calculate the downstream prediction $\hat{\mathbf{y}} = (g_t \circ h_t^S)(\mathbf{x})$
        Update the parameters of $h_t^S$ and $g_t$ to minimize the downstream loss $\mathcal{L}_{ds}(\hat{\mathbf{y}}, \mathbf{y})$
    **end for**
    The student becomes the teacher for the next generation: $h_{t+1}^T \leftarrow h_t^S$
**end for**
Return the last (or the best) $(g_t \circ h_t^S)(\mathbf{x})$ for the downstream task

---

# A  The Ladder of Compositionality

Table 3: What information contained in $\mathbf{z}$ on the ladder and their corresponding capabilities. $H(\cdot)$ is the entropy, $\leftrightarrow$ means bijection, $\Leftrightarrow$ means isomorphism bijection.

| | Infor. in $\mathbf{z}$ | Train acc. | ID-gen | OOD-gen with seen concepts | Comp-gen |
|---|---|---|---|---|---|
| Stage I | $H(\mathbf{G} \mid \mathbf{z}) > 0$ | ✗ | ✗ | ✗ | ✗ |
| Stage II | $H(\mathbf{G} \mid \mathbf{z}) = 0$ $H(\mathbf{z} \mid \mathbf{G}) > 0$ | ✓ | ✓ | ✗ | ✗ |
| Stage III | $\mathbf{z} \leftrightarrow \mathbf{G}$ | ✓ | ✓ | ✓ | ✗ |
| Stage IV | $\mathbf{z} \Leftrightarrow \mathbf{G}$ | ✓ | ✓ | ✓ | ✓ |

To figure out what $\mathbf{z}$ we need in order to generalize well compositionally, we propose the "ladder of compositionality" in Figure 1. To justify our claims, this appendix will discuss how we could climb the ladder step by step by analyzing how the corresponding requirements are generated. A formal definition of the compositional mapping in terms of group theory, which is necessary for reaching the final stage of the ladder, is also provided. In short, we find only relying on the mutual information between $\mathbf{z}$ and $\mathbf{G}$ (or between $\mathbf{x}$ and $\mathbf{y}$) cannot reach the final rung of the ladder: we need other inductive biases, which is the main motivation of this paper.

## A.1  Stage I: z misses some important information in G

The learned representation would have $H(\mathbf{G} \mid \mathbf{z}) > 0$ at this stage. As $Y = \mathsf{GenY}(\mathbf{G}, \epsilon)$ and $\hat{Y} = g(\mathbf{z})$ are assumed to be invertible (here $Y$ and $\hat{Y}$ are random variables), this condition can be rewritten as $I(Y; \hat{Y}) < H(Y)$ [6]. Hence following the analysis in [69], a model with such an encoder $\mathbf{z} = h(\mathbf{x})$ even cannot achieve high enough training performance, let alone generalizing to unseen

---

[5]In practice, we can choose to randomly initialize the speaker (usually when the model is small), copy the pretrained checkpoint (when the model is large), or copy the parameters of the teacher in previous generations (the seed iterated learning variant mentioned in [50]).

[6]Using the fact that $H(\mathbf{G} \mid \mathbf{z}) = H(Y \mid \hat{Y}) = H(Y) - I(Y; \hat{Y})$.

test distributions. This condition might occur when the model underfits the training data, e.g., at the beginning of training or the model's capacity is too small. To make an improvement, one can increase the model size or train longer.

## A.2 Stage II: z not only contains all information in G but also some in O

At this stage, the learned representation would have $H(\mathbf{G} \mid \mathbf{z}) = 0$ and $H(\mathbf{z} \mid \mathbf{G}) > 0$. This means $h(\mathbf{x})$ remembers additional information that $\mathbf{G}$ doesn't have, e.g., noisy information in $\mathbf{O}$. From $H(\mathbf{G} \mid \mathbf{z}) = 0$, we know $H(Y \mid \hat{Y}) = 0$ and hence $I(Y; \hat{Y}) = H(Y)$. Then the model would have perfect training performance and could also be able to generalize well when training and test datasets share the same distribution. However, when facing the out-of-distribution generalization problem, especially when a spurious correlation exists between some factors in $\mathbf{O}$ and $\mathbf{G}$, the extra information learned by $\mathbf{z}$ can mess the predictions up. Such a phenomenon is named "short-cut learning" and is quite common in many deep-learning systems [20]. For example, if the background strongly correlates with the object in the training set (e.g., a cow usually co-occurs with the grass while a seagull usually co-occurs with the beach), the DNN then tends to rely more on these "short-cut" features (e.g., the background rather than the object) during training. If such correlations disappear or reverse in the test set, the models relying on factors in $\mathbf{O}$ cannot generalize well to a new distribution.

**Making improvement – loss based on information-bottleneck**    To make an improvement, the model should eliminate the task-irrelevant information as much as possible. Based on this principle, authors of [74] propose to minimize the following information bottleneck equations:

$$\max \quad I(Z; Y) - \beta I(Z; X), \quad \beta > 0, \tag{3}$$

which means the learned $\mathbf{z}$ should extract as much as information from $Y$ (or equivalently, $\mathbf{G}$) and forget as much as irrelevant information about $X$ (i.e., those in $\mathbf{O}$). This method is also widely applied in other relevant tasks, like domain adaptation [46], invariant risk minimization [1], and etc.

**Making improvement – data augmentation**    Another simple and efficient way to make improvements is data augmentation: one can identify some task-irrelevant factors in $\mathbf{O}$ and design specific data augmentation methods to teach the model to be insensitive to them. For example, if we believe that the label of an image should be irrelevant to color jittering, random cropping, rotation, flipping, etc., we can apply random augmentations during training and treat the differently augmented $\mathbf{x}$ and $\mathbf{x}'$ as the same class. Then the model would inherently learn to be insensitive to such factors and hence forget the corresponding information. One interesting thing about data augmentation is that it can be designed and applied in a reverse direction, i.e., we can break some semantic factors and train the model to be insensitive to the broken samples. For example, believing the shape of the image and order of the words are semantic factors in $\mathbf{G}$, the authors of [56] propose to randomly rotate the image patches or words to make negative samples. Those models that perform well on such negative samples are more likely to rely on the factors in $\mathbf{O}$.

**Making improvement – auxiliary task design, e.g., SSL**    Furthermore, one can also consider designing auxiliary tasks in addition to the downstream task, e.g., pretrain using self-supervised learning (SSL) and finetune on the target task. In [67], the authors empirically show that the representations learned via SSL usually generalize better than those learned via supervised learning when facing OOD downstream problems, even though the models are trained using a similar amount of data samples. There are also some works demonstrating that SSL representations encode more semantic information about the input image [10], which is a sign that auxiliary tasks like SSL can introduce extra biases that favor information in $\mathbf{G}$.

Consider the first group of SSL methods, which are usually based on contrastive loss, e.g., SimCLR [11], MoCo [30], etc. These methods usually require $h(\mathbf{x}_i)$ and $h(\mathbf{x}_i')$ to be similar while $h(\mathbf{x}_i)$ and $h(\mathbf{x}_j), i \neq j$ to be distinct, where $\mathbf{x}_i$ is the anchor input, $\mathbf{x}_i'$ is the augmentation of it, and $\mathbf{x}_j$ is another different image. The carefully designed augmentation can encourage the model to ignore some task-irrelevant factors that belong to $\mathbf{O}$. Imagine $\mathbf{x}_i'$ is generated by deleting the background of $\mathbf{x}_i$. As the training enforces $d_z(h(\mathbf{x}), h(\mathbf{x}'))$ to be small, the learned model will then become insensitive to the information in the background, and hence avoid relying on this "short-cut" feature. Note that the contrastive SSL is utilizing the bias from data augmentation in a more aggressive way: the SSL algorithm will tell the model that $\mathbf{x}$ and $\mathbf{x}'$ are the *same image* while the supervised learning only inform the model that $\mathbf{x}$ and $\mathbf{x}'$ belong to the *same class*.

Another line of SSL is built on reconstruction tasks, like denoising auto-encoder (DAE [77]) and masked auto-encoder (MAE [29]). The $h(\mathbf{x})$ in these methods is usually trained in an auto-encoder fashion: using a reconstruction network $r(\mathbf{z}) : \mathcal{Z} \to \mathcal{X}$, we require the reconstructed $\mathbf{x}_{recon} = (r \circ h)(\mathbf{x})$ to be similar to the original input $\mathbf{x}$. As the above equation has a trivial solution, i.e., $(r \circ h)(\cdot) = identity$, which means $h(\mathbf{x})$ copies every details about $\mathbf{x}$ (including all $\mathbf{O}$ and $\mathbf{G}$), some early works like denoising auto-encoder propose to add noise on $\mathbf{x}$ to encourage non-trivial solutions. Depending on the noise we introduce, the model will learn to ignore different factors accordingly, which seems quite similar to the data augmentation mentioned in contrastive SSL methods. To encourage $h(\mathbf{x})$ to extract more useful semantic information in $\mathbf{G}$, methods like MAE [29] propose to mask most of the patches of the input image and try to make reconstructions based on the remaining patches. Such methods exhibit amazing reconstruction performance (not in terms of high resolution, but the precise semantic reconstruction), which also implies that the $h(\mathbf{x})$ trained in this way is capable of extracting high-level semantic generating factors (those are likely in $\mathbf{G}$). Furthermore, methods like BEiT [4] and iBOT [83] also patchify and mask the input images, and concurrently, impose extra constraints on $\mathbf{z}$ by comparing them with the descriptions generated by the big language model. Such designs also encourages $h(\mathbf{x})$ to extract high-level semantic information, as illustrated in [10].

In summary, as the SSL algorithms learn good $\mathbf{z}$ by designing loss or tasks on it, we can introduce extra inductive bias via auxiliary tasks. By requiring $\mathbf{z}$ to be invariant when adding noise or conducting data augmentation on $\mathbf{x}$, the task-irrelevant information can be ruled out during learning. By requiring $\mathbf{z}$ contains the necessary information for reconstruction when only part of $\mathbf{x}$ is observable, the task-relevant semantic information can be highlighted during learning. By combining these principles, it is possible for us to learn good $h(\mathbf{x})$ that only extracts information in $\mathbf{G}$. With the help of these methods, our $\mathbf{z}$ might learn *exactly* all information in $\mathbf{G}$, which means the third rung is achieved.

### A.3 Stage III: z learns exactly all information in G, i.e., $h(\cdot)$ leads to a bijection

Starting from Stage II, if we can design clever training methods (e.g., adding regularization, data augmentation, auxiliary task, etc.) and make our $h(\mathbf{x})$ to be insensitive to some $\mathbf{O}$, ideally, we can learn an almost perfect encoder that extracts exactly all information contained in $\mathbf{G}$. In this case, we have $H(\mathbf{G} \mid \mathbf{z}) = 0$ and $H(\mathbf{z} \mid \mathbf{G}) = 0$, i.e., $h(\cdot)$ leads to a bijection between $\mathbf{z}$-space and $\mathbf{G}$-space, which is denoted as $\mathbf{z} \leftrightarrow \mathbf{G}$. Ideally, such $\mathbf{z}$ can generalize well even when $P_{train} \neq P_{test}$, as long as all the concepts in the test set are seen by $h(\mathbf{x})$ during training, i.e., $\mathsf{supp}[P_{test}(\mathbf{G})] \subseteq \mathsf{supp}[P_{train}(\mathbf{G})]$. However, we speculate that even $h(\mathbf{x})$ on Stage III will struggle in the comp-gen problem, as the problem assumes $\mathsf{supp}[P_{test}(\mathbf{G})] \cap \mathsf{supp}[P_{train}(\mathbf{G})] = \emptyset$. In other words, we need the model to decompose and recombine the learned concepts in a systematic way, which is similar to the ground-truth-generating mechanism. To achieve this goal, we need to consider how to achieve Stage IV.

### A.4 Stage IV: $h(\cdot)$ leads to a isomorphism bijection between z and G

To generalize well compositionally, we not only need $\mathbf{z}$ contains exactly all information in $\mathbf{G}$, the structure of $\mathbf{G}$ should also be embodied in $\mathbf{z}$, which means $h(\cdot)$ should lead to an isomorphism bijection between $\mathbf{z}$-space and $\mathbf{G}$-space (i.e., $\mathbf{z} \Leftrightarrow \mathbf{G}$). Specifically, we need:

**Hypothesis 1 in detail.** To generalize well compositionally, we need $\mathbf{z} \Leftrightarrow \mathbf{G}$, which requires:

1. $\mathbf{z} \leftrightarrow \mathbf{G}$, i.e., $h(\cdot)$ leads to a bijection between $\mathbf{z}$ and $\mathbf{G}$;

2. $P(\mathbf{z})$ and $P(\mathbf{G})$ factorize in a similar way;

3. Each $z_i$ maps to some $G_{\mathbf{w}_i}$, where $\mathbf{w}$ is a permutation vector of length $m$, and such a mapping is invariant[7];

4. For each $z_i$, the mapping between $z_i = k$ and $G_{\mathbf{w}_i} = \mathbf{u}_k$ is invariant[8], where $\mathbf{u}$ is a permutation vector of length $v_{\mathbf{w}_i}$.

We call such mappings **compositional** and call the other bijections **holistic**. To better understand the difference between these two types of mappings, we consider a simple example where the generating factors are $\mathbf{G} = [G_1, G_2]$, $G_1 = \{\text{blue, red}\}$ and $G_2 = \{\text{circle, box}\}$, and the representations are

---

[7]For example, $z_1$ always encode color and $z_2$ always encode shape.

[8]For example, $z_1 = 0$ always means blue and $z_1 = 1$ always means red.

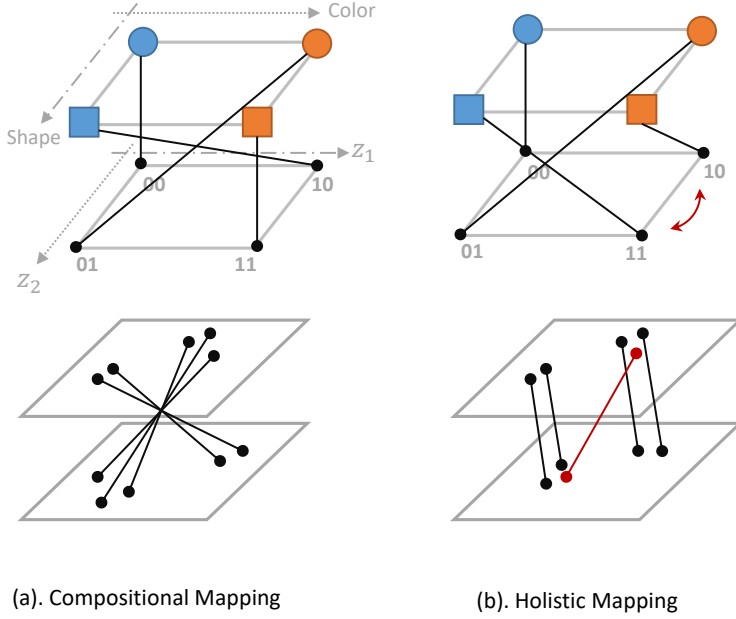

(a). Compositional Mapping    (b). Holistic Mapping

Figure 5: A compositional mapping and a holistic mapping. Although holistic mapping in this example seems to have fewer "crossings", it has lower topological similarity.

Table 4: An example of coding the mappings, where $\alpha$ is how many characters (including space and unique symbol, e.g., $\rightarrow$ and :) are used to express the grammar.

| A compositional mapping 5 rules, $\alpha = 43$ | | | A holistic mapping 4 rules, $\alpha = 56$ | | |
|---|---|---|---|---|---|
| S | $\rightarrow$ | z2, z1 | | | |
| z2: | 0 | $\rightarrow$ blue | S: | $\rightarrow$ | blue circle |
| z2: | 1 | $\rightarrow$ red | S: | $\rightarrow$ | red circle |
| z1: | 0 | $\rightarrow$ circle | S: | $\rightarrow$ | red box |
| z1: | 1 | $\rightarrow$ box | S: | $\rightarrow$ | blue box |

$\mathbf{z} = [z_1, z_2]$, $z_1 = \{0, 1\}$ and $z_2 = \{0, 1\}$. Hence the space of $\mathbf{G}$ is { blue circle, blue box, red circle, red box }, and the space of $\mathbf{z}$ is $\{00, 01, 10, 11\}$. To generate a compositional mapping, we can first decide the meaning of $z_i$, e.g., $z_1$ represents the shape and $z_2$ represents the color. After that, we assign vocabularies in $z_i$ to denote different meanings, e.g., $z_1 = 0 \rightarrow$ circle, $z_1 = 1 \rightarrow$ box, $z_2 = 0 \rightarrow$ blue, and $z_2 = 1 \rightarrow$ red. Combining these two steps, we generate a compositional mapping, as illustrated in Figure 5(a). Any bijections that cannot be decomposed in this way are holistic mappings, like Figure 5(b). Obviously, the compositional mappings can generalize compositionally while the holistic ones cannot. However, finding a compositional mapping from all possible bijections is a hard problem, as the number of compositional mappings is much smaller than the holistic ones: assuming we have $m = 3$ different $G_i$ and each with $v = 3$ possible values, there are roughly $10^3$ compositional mappings ($m!(v!)^m$), and $10^{28}$ holistic mappings ($(v^m)! - m!(v!)^m$).

Another important characteristic of compositional mappings is the "distance-preserving" ability. In other words, they will map two $\mathbf{x}$ with similar $\mathbf{G}$ to two similar $\mathbf{z}$ in the representation space, i.e., $\mathbf{z}$ preserve the topological structure of $\mathbf{G}$. We can use topological similarity ($\rho$, defined in Equation 2) to quantify how compositional a mapping is. Usually, for all bijections, mappings with larger $\rho$ are more compositional.

To sum up the aforementioned four stages of the learned $\mathbf{z}$, we list the conditions that $\mathbf{z}$ must satisfy and their capabilities in Table 3.

### A.5 Relationship with disentangled representations

Readers might notice that our problem settings and the requirements for **z** mentioned in Hypothesis 1 are quite similar to those discussed in disentanglement representation learning [32]. Here we explain the connections and differences between them.

**We care more about downstream tasks.** In disentanglement representation learning, people focus more on a general property of the learned representation by assuming the existence of independent semantic generating factors. Disentanglement is then considered a desired property of such representations. Works in this line try to formalize this property and believe this property is beneficial for multiple downstream tasks. However, as the downstream task is not incorporated in such analysis, it is hard to conclude whether specific factors are semantic or not (remember the split of **O** and **G** highly depends on the task). On the other hand, this paper focuses more on the generalization ability of the learned representations on specific downstream task(s). We directly optimize the loss of the downstream task in the interaction phase, which we believe can regularize the representations more efficiently.

**We don't strictly require a disentangled z.** Although representations with properties like consistency or restrictiveness mentioned in [68] could be beneficial for the downstream tasks, it seems not necessary to have all of them. In other words, disentanglement is a sufficient but not necessary requirement for a model to generalize well in downstream tasks. That is because the factorization of **G** could be non-trivial (as nature is not simple). We believe that capturing the hidden structure of **G** and $\mathsf{GenY}(\cdot)$ using **z** and $g(\cdot)$ is more important than mapping an involved generating mechanism to a disentangled system. Additionally, implicitly splitting **O** from **G** is rather crucial in our settings, which is rarely discussed in the fields of disentanglement representation learning.

**We mainly consider discrete factors.** Inspired by how human language evolves, it is natural to start from the discrete factors and representation due to the discreteness of human language. Our experimental results also benefit a lot from the discreteness, e.g., using cross-entropy loss to amplify the learning speed advantage, sampling pseudo labels to strengthen the inductive bias, using group theory to formalize the compressibility and Kolmogorov complexity, etc. However, as nature might not be purely discrete, incorporating the continuous latent space is crucial to enlarge the scope of our study. We would leave this in our future work.

## B Compositionality, Compressibility, Kolmogorov complexity, and number of active bases

This appendix links several key concepts related to compositional mappings together, i.e., compressibility, Kolmogorov complexity, and number of active bases. The analysis here provides good intuition on why we might expect iterated learning to be helpful in comp-gen.

### B.1 Higher compositionality, lower Kolmogorov complexity

We first complete the proof of Proposition 1.

**Proposition 1** (Informal). *For $m, v \geq 2$, among all bijections, any compositional mapping has much lower Kolmogorov complexity than a typical non-compositional mapping.*

*Proof.* Recall the fact that any bijection from **z** to **G** can be represented by an element in the symmetry group $S_{v^m}$. From the definition of the symmetry group, we know each element in $S_{v^m}$ can be represented by a permutation matrix of size $v^m$. As there is only one $1$ in each row and column of a permutation matrix, any permutation matrix can be uniquely represented by a permuted sequence of length $v^m$. Specifically, assume we have a sequence of natural numbers $\{1, 2, ..., v^m\}$, each permuted sequence $\mathsf{Perm}(\{1, 2, ..., v^m\})$ represents a distinct permutation matrix, and hence represents a distinct bijection from **z** to **G**. In other words, we can encode one bijection from **z** to **G** using a sequence of length $v^m$, i.e., $\mathsf{Perm}(\{1, 2, ..., v^m\})$, and bound the corresponding Kolmogorov complexity (in bits) as

$$\mathcal{K}(\text{bijection}) \leq v^m \cdot \log_2 v^m = v^m \cdot m \cdot \log_2 v, \tag{4}$$

As an arbitrary bijection from **z** to **G** doesn't have any extra information to improve the coding efficiency, Equation (4) provides an upper bound of the minimal Kolmogorov complexity.

On the contrary, as each compositional mapping can be represented by an element in $S_v^m \rtimes S_m$, we can encode the mappings more efficiently. Specifically, we need to first use $m$ sequences with length $v$, i.e., $\mathsf{Perm}(\{1, 2, ..., v\})$, to represent the assignment of "words" for each $z_i$. After that, we need one sequence of length $m$, i.e., $\mathsf{Perm}(\{1, 2, ..., m\})$ to encode the assignment between $z_i$ and $G_j$. The corresponding Kolmogorov complexity is then bounded as

$$\mathcal{K}(\text{comp}) \leq v \cdot \log_2 v + m \cdot \log_2 m, \tag{5}$$

Although this is only an upper bound, by a counting argument *most* such mappings must have a complexity no less than, say, a constant multiple of that bound.

To compare the Kolmogorov complexity, we can define a ratio as $\gamma \triangleq \frac{\mathcal{K}(\text{bijection})}{\mathcal{K}(\text{comp})}$. Obviously, when $m \leq v, \gamma \geq \frac{v^{m-1} \cdot m}{2}$, which is larger than 1 as long as $m, v \geq 2$. When $m > v, \gamma \geq \frac{v^m \log_2 v}{2 \log_2 m}$, which is also larger than 1 when $m, v \geq 2$. $\qquad\square$

Actually, there might be some mappings that are not purely compositional or holistic. For example, we can have a mapping with $z_{i \leq 10}$ sharing the reused rules while other $z_{i > 10}$ doesn't. Then this type of mapping can be represented by an element in $S_v^{10} \rtimes S_{10} \rtimes S_{v^{m-10}}$. As a mapping in this subset shares 10 common rules, its Kolmogorov complexity is between $\mathcal{K}(\text{bijection})$ and $\mathcal{K}(\text{comp})$. Intuitively, *for all bijections*, smaller $\mathcal{K}(\cdot)$ means higher compressibility and higher compositionality.

## B.2 Regularize the Kolmogorov complexity using iterated learning

From the analysis in Appendix A.4 and B.1, we know that finding mappings with lower Kolmogorov complexity is the key to generalizing well compositionally. From existing works in cognitive science, we know iteratively introducing new agents to learn from old agents can impose the compressibility pressure and hence make the dominant mapping more compact after several generations [42]. Although iterated learning reliably prompts the emergence of compositional mapping in lab experiments, directly applying it to deep learning is not trivial: as we are not sure whether the preference for compositionality still exists for the neural agents. Hence in this subsection, we study a simple overparameterized linear model on a 0/1 classification task to show that iterated learning can indeed introduce a non-trivial regularizing effect. Combining with the fact that mappings with lower Kolmogorov complexity are more likely to capture the ground truth generating mechanism and hence generalize better [71], we can conclude that iterated learning is helpful for compositional generalization problems.

Consider a general supervised learning problem, in which we want to learn a mapping $f \in \mathcal{F} : \mathcal{X} \to \mathcal{Y}$ that could approximate the underlying relationship between random variables $X$ and $Y$. As we usually have a finite number of training samples and the space of all possible mappings is large, the model could just remember all $(\mathbf{x}, y)$ pairs in the training set to achieve a perfect training performance. To avoid this trivial solution, we usually expect the optimal $f^*$ to have specific properties, e.g., smoothness or Lipschitz continuousness, etc. Hence usually, we want to optimize a problem with a corresponding regularization term:

$$f^* \triangleq \arg \min_{f \in \mathcal{F}} R(f) \quad \text{s.t.} \quad \frac{1}{N} \sum_n \|f(\mathbf{x}_n) - y_n\|_2^2 \leq \epsilon, \tag{6}$$

where $R : \mathcal{F} \to \mathbb{R}$ is regularizing $f$ and $\epsilon$ is the training loss tolerance. This regularization term is usually the inner product of $f$ on the functional space, i.e., $R(f) = \|f\|_{\mathcal{H}}$, where $\mathcal{H}$ is a reproducing kernel Hilbert space (RKHS) determined by some kernel function $\kappa(\cdot, \cdot)$. For example, if we consider $\forall (x, y) \in [0, 1]^2$ and $\kappa(x, y) = \min(x, y)$, then $\|f\|_{\mathcal{H}} = \|f'\|_{[0,1]^2}$ [64]. In other words, the regularizer will penalize functions with higher first-order derivatives.

In order to make the analysis generalize to other properties of $f$, we define a linear differential operator $L$ as $[Lf] \triangleq \int_{\mathcal{X}} u(\mathbf{x}, \cdot) f(\mathbf{x}) \, d\mathbf{x}$, where $u(\cdot, \cdot)$ is a kernel function. Then, the regularization term is:

$$R(f) = \|f\|_{\mathcal{H}} = \langle f, f \rangle_{\mathcal{H}} = \langle Lf, Lf \rangle_{\mathcal{X}^2} = \int_{\mathcal{X}} \int_{\mathcal{X}} u(\mathbf{x}, \mathbf{x}^\dagger) f(\mathbf{x}) f(\mathbf{x}^\dagger) \, d\mathbf{x} \, d\mathbf{x}^\dagger \tag{7}$$

Substituting this definition back to Equation 6 and then applying the Karush-Kuhn-Tucker (KKT) conditions, the closed-form solution for this optimization problem is (i.e., Proposition 1 in [53]):

$$f^*(\mathbf{x}) = g_{\mathbf{x}}^\top (cI + G)^{-1}\mathbf{Y}, \tag{8}$$

where $c$ is a bounded constant, $\mathbf{Y} = [y_1|\ldots|y_N]^\top$ is the stacked training labels. The matrix $G \in \mathbb{R}^{N \times N}$ and its vector $g_{\mathbf{x}} \in \mathbb{R}^{N \times 1}$ is defined as:

$$G[j, k] \triangleq \frac{1}{N}g(\mathbf{x}_j, \mathbf{x}_k); \qquad g_{\mathbf{x}}[k] \triangleq \frac{1}{N}g(\mathbf{x}, \mathbf{x}_k). \tag{9}$$

The $g(\mathbf{x}, \mathbf{q})$ is the Green's function [63] of this operator and is defined by:

$$\int_{\mathcal{X}} u(\mathbf{x}, \mathbf{x}^\dagger)g(\mathbf{x}, \mathbf{q})\, \mathrm{d}\mathbf{x}^\dagger = \delta(\mathbf{x} - \mathbf{q}), \tag{10}$$

where $\delta$ is the Dirac delta function. Following the definition of Green's function, we know $G$ is positive definite and hence decompose it as:

$$G = V^\top DV, \tag{11}$$

where $D = \mathrm{diag}([d_1, \ldots, d_N])$ is determined by its eigenvalues and $V$ contains $N$ corresponding eigenvectors. Now, we can stack the model's prediction for different input samples $\mathbf{x}_n$ and get the vector form solution of problem 6:

$$\mathbf{f}^* \triangleq [f^*(\mathbf{x}_1)|\cdots|f^*(\mathbf{x}_k)]^\top = G^\top(cI + G)^{-1}\mathbf{Y} = V^\top D(cI + G)^{-1}V\mathbf{Y} \tag{12}$$

With this solution, following the settings in [53], we can explain where the compressibility pressure (the one that favors mappings with lower Kolmogorov complexity) comes from. Specifically, the optimal model for the first generation is $\mathbf{f}_0^* = V^\top D(c_0 I + G)^{-1}V\mathbf{Y}_0$. Then in the following generations, the model in generation $t$ will learn from the predictions of the model in the previous generation. As the problem is identical (the only difference is the labels) for different generalizations, we can have the following recursion formulas:

$$\mathbf{f}_t^* = V^\top D(c_t I + G)^{-1}V\mathbf{Y}_t \quad \text{and} \quad \mathbf{Y}_t = \mathbf{f}_{t-1}^*. \tag{13}$$

Solving this yields the expression of the labels in the $t$-th generation:

$$\mathbf{Y}_t = V^\top A_{t-1}V\mathbf{Y}_{t-1} = V^\top \left(\prod_{i=0}^{t-1} A_i\right)V\mathbf{Y}_0, \tag{14}$$

where $A_t \triangleq D(c_t I + D)^{-1}$ is a $N \times N$ diagonal matrix. Substituting this back to Equation 12, we finally obtain the following expression:

$$f_t^*(\mathbf{x}) = g_{\mathbf{x}}^\top V^\top D^{-1}\left(\prod_{i=0}^{t} A_i\right)V\mathbf{Y}_0 \tag{15}$$

$$\mathbf{f}_t^* = GV^\top D^{-1}\left(\prod_{i=0}^{t} A_i\right)V\mathbf{Y}_0$$

$$= V^\top \left(\prod_{i=0}^{t} A_i\right)V\mathbf{Y}_0. \tag{16}$$

From this solution, the model's prediction at $t$-th generation can be considered as a weighted combination of transformed $\mathbf{Y}_0$. The matrix $V$ will first map $\mathbf{Y}_0$ to a space determined by the Green's function. Different dimensions of this space are then rescaled by a diagonal matrix $\prod_{i=0}^{t} A_i$. After that, the vector is transformed back to the origin space by multiplying $V^\top$. Among these terms, $\prod_{i=0}^{t} A_i$ is the only one that depends on $t$. Recall the definition of $A_t = D(c_t I + D)^{-1}$, we can conclude that $\prod_{i=0}^{t} A_i$ is also a diagonal matrix where each entry has the form like $\prod_t \frac{d_j}{c_t+d_j}$ (here $d_j$ is the $j$-th eigenvalue of $G$). As $c_t > 0, \forall t$, as stated in [53], all the diagonal entries will gradually decrease when $t$ grows. The dimensions with smaller $\prod_t \frac{d_j}{c_t+d_j}$ decrease faster, and vice versa. Recall

the role played by this diagonal matrix, we can imagine that the number of active bases in $V$ is decreasing when $t$ grows, which is a strong and unique inductive bias (i.e., compressibility pressure) introduced by this recursive training fashion[9].

Now, we can link these theoretical analyses to the Kolmogorov complexity and compressibility pressure. The crux is the understanding of "active bases". Consider a toy example where $\mathbf{G} = [G_1, G_2]$, where each $G_i$ has 4 possible values (there are $N = 16$ different objects). Then, to memorize these 16 samples, the model needs 16 bases like "S 00 $\rightarrow$ blue circle". However, for compositional mappings, only 9 bases are enough[10]: because the model *reuse* some rules.

In summary, when $t$ is small, there is no preference for compositional mappings because the number of active bases is large enough to remember most of the training samples. As $t$ increases, the model then needs to be clever enough to reuse some bases, where the structure of the mapping emerges. If $t$ is too large, where the compressibility pressure is too strong, the model will degenerate into a very naive solution, which is harmful for generalization (hence we need the interaction phase in iterated learning, discussed later).

# C  Experiments on Controlled Vision dataset

## C.1  Experimental Settings

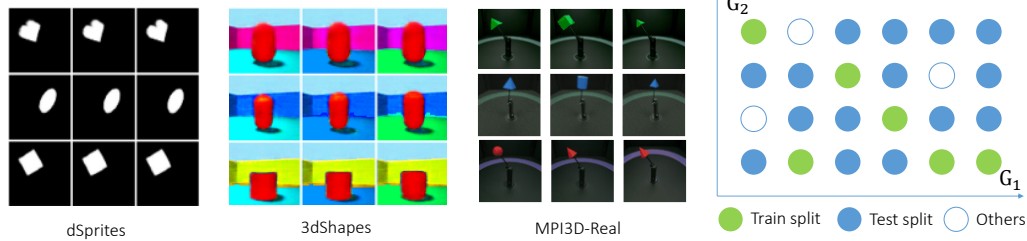

Figure 6: The toy vision datasets and a train/test split example. All images have size 64*64. Pixels in dSprite are binary values while those in 3dShape and MPI3D contain 3 channels.

Table 5: Vision datasets considered in this paper. The numbers in the parentheses represent how many different values of the attribute. The last column means how many different samples we select for each $\mathbf{G}$. Hence the number of samples in both training and test sets of these three datasets would be 9000, 8000, and 7200.

| | $G_1$ | $G_2$ | $G_3$ | $G_4$ | $|\mathcal{G}|$ | $\mathbf{O}$ | # per $\mathbf{G}$ |
|---|---|---|---|---|---|---|---|
| dSprites | shape (3) | scale (6) | pos-x (10 out of 32) | pos-y (10 out of 32) | 1600 | orientation (40) | 5 |
| 3dShape | floor hue (10) | wall hue (10) | object hue (10) | object scale (8) | 8000 | object shape (4) orientation (15) | 1 |
| MPI3D | object color (6) | object shape (6) | horizontal x (10 out of 40) | vertical y (10 out of 40) | 3600 | size (2) camera (3) background (3) | 2 |

**Data generating factors**  In this paper, we conduct experiments on three vision datasets, i.e., dSprites [52], 3dShapes [9], MPI3D-real [23], where the ground truth $\mathbf{G}$ are given. The summary and examples of these datasets are provided in Table 5 and Figure 6.

Here we specify how to split the dataset and generate the downstream labels using 3dShapes as an example. We denote the hue of the floor, wall, and object as $G_1, G_2,$ and $G_3$, respectively; each has 10 possible values, linearly spaced in $[0, 1]$. The object scale, $G_4$, has 8 possible values linearly spaced in $[0, 1]$. The remaining two factors, object shape (4 possible values) and object orientation (15 possible values), are treated as *other factors* and merged into $\mathbf{O}$. Data augmentation methods such as adding Gaussian noise, random flipping, and so on, are also merged into $\mathbf{O}$. Under this setting, the universe of $\mathbf{G}$, i.e., $\mathcal{G}$, has 8000 different values, which is further divided into $\mathcal{G}_{train}$ and $\mathcal{G}_{test}$. For

---

[9]The authors in [53] also prove that such a regularization cannot be achieved by other forms of regularizations.

[10]Specifically, we need one basis like "S $\rightarrow$ z1 z2", four bases like "z0 i $\rightarrow$ some color", and four bases like "z1 j $\rightarrow$ some shape". Please refer to Table 4.

the sys-gen problem, we assume $\mathcal{G}_{train} \cap \mathcal{G}_{test} = \emptyset$. One measure of the difficulty of the problem is the split ratio, $\alpha = |\mathcal{G}_{train}|/|\mathcal{G}|$; smaller $\alpha$ generally means a more challenging problem.

**Data generating mechanisms** For the training set, we first select $\mathbf{G} \in \mathcal{G}_{train}$, and then generate multiple input signals using $\mathbf{x} = \mathsf{GenX}(\mathbf{G}, \mathbf{O})$, where $\mathbf{O}$ is uniformly random. The $\mathbf{x}$ in the test set is generated in a similar way, but without including data augmentation in $\mathbf{O}$. Labels for all pairs in the dataset are generated by $\mathbf{y} = \mathsf{GenY}(\mathbf{G}, \epsilon)$. The main downstream task we study here is regression: $\mathsf{GenY}(\mathbf{G}, \epsilon) = \mathbf{a}^\top \mathbf{G} + \epsilon$, where $\mathbf{a} = [a_1, ..., a_m]^\top$ is a column vector and all $a_i$ are chosen from $[0, 1]$[11]. These examples assume $\mathbf{y}$ is generated by a simple combination of different $G_i$, where recovering the generating factors is necessary to generalize well compositionally.

**Model and training settings** The model structure for this section is illustrated in Figure 2. We consider using a randomly initialized 4-layer CNN for dSprites and ReNet18 [31] for 3dShapes and MPI3D. Unless otherwise specified, we consider a linear head $g(\mathbf{z})$, and a typical $\mathcal{L}_{ds}$, i.e., cross-entropy loss for classification and mean square error loss for regression. The networks are optimized using a standard SGD optimizer with a learning rate of $10^{-3}$ and a weight decay rate of $5 * 10^{-4}$. Actually, we find the results are insensitive to these settings.

## C.2  More Results

### C.2.1  Some interesting observations

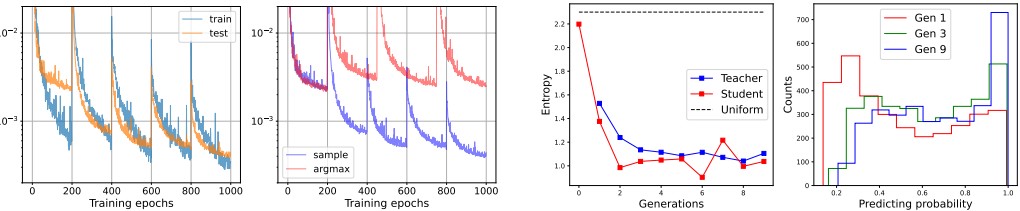

Figure 7: Left to right: 1.) zoom in on the curves of training and test loss; 2.) demonstration of what will happen to test loss when replacing the sampling by argmax in the imitation phase; 3.) how the average predicting entropy, i.e., $\mathbb{E}[H(\bar{z}_l)]$, changes; 4.) histograms of predicting probabilities in different generations. All panels come from experiments of SEM-IL on 3dShapes.

See the first panel in Figure 7, which demonstrates the training and testing loss when training a model using SEM-IL. In the first generation, we see the training loss is always smaller than the test one. The test loss then plateaus after some epochs, which matches our expectations. However, in the following generations, the test loss would decrease faster than the training one at the beginning of the interaction phase, which is quite counter-intuitive. We would explore why this happens and whether it is a sign of increased topological similarity in the future.

Another observation is about the sampling mechanism applied in the imitation phase. Remember in Algorithm 1, the pseudo labels used in the imitation phase are sampled from the teacher's prediction $\bar{z}_l$. Hence if the teacher is confident in some attributes, the generated labels would be consistent in different epochs, and vice versa. From Figure 4, we provide the scatter plot of the correlation between the teacher's confidence and the student's learning speed. Here we verify this hypothesis through an ablation study. Specifically, we replace the sampling procedure with an argmax function, i.e., the teacher always provides the label with the largest predicting probability regardless of its confidence. As illustrated in the second panel in Figure 7, the test performance of the argmax-case is much worse than the standard SEM-IL method.

The benefits introduced by sampling pseudo labels can also be interpreted as we are making self-adapting $\tau$ for different input samples in SEM. In the origin SEM, we only have one $\tau$ to control the average entropy of the backbone's prediction (lower $\tau$ leads to peakier predicting distributions, and vice versa). However, a model trained using SEM-IL equivalently has different $\tau$ for different $\mathbf{x}$.

---

[11]We also tried a simple non-linear mapping from $\mathbf{G}$ to $y$, i.e., $\mathsf{GenY}(\mathbf{G}, \epsilon) = a_1 \cdot G_1 + a_2 \cdot G_2 + a_3 \cdot G_4 G_3 + \epsilon$, and a multi-task scenario, where $\mathbf{a} \in \mathbb{R}^{|y| \times m}$ is a matrix. The resulting trends in these settings are quite similar.

To understand this, we can compare the entropy of $h_1(\mathbf{x})$ and $h_1(\mathbf{x}')$, where the teacher is confident in $(\mathbf{x}, \mathbf{g}_1 = 0)$ and less confident in $(\mathbf{x}', \mathbf{g}_1 = 0)$. Then during imitation, the student will remember $(\mathbf{x}, \mathbf{g}_1 = 0)$ much faster than $(\mathbf{x}', \mathbf{g}_1 = 0)$ and hence assign $\bar{z}_1 = 0$ higher probability when the input is $\mathbf{x}$. On the contrary, for the input $\mathbf{x}'$, as the student might receive different corresponding pseudo labels during imitation, it would assign a lower probability for $\bar{z}_1 = 0$ given $\mathbf{x}'$. As a result, the entropy $H(\bar{z}_1 \mid \mathbf{x})$ and $H(\bar{z}_1 \mid \mathbf{x}')$ of the student model would be very different as the teacher have different confidence when generating the pseudo labels, which is equivalent as automatically selecting different $\tau$ for $\mathbf{x}$ and $\mathbf{x}'$.

The last two panels in Figure 7 demonstrate how the entropy of the model's prediction on $\mathbf{G}$ changes in different generations. (Remember the output of the backbone after SEM, i.e., $\mathbf{z} = [\bar{z}_1, \ldots, \bar{z}_m]$, are $m$ simplicial vectors with length $v$.) From the figures, we see the entropy gradually decreases as the training goes on, which means the model is becoming more and more confident in its predictions on average. However, the last panel shows that there are still many unconfident predictions even after converging on the downstream task performance: we speculate that these factors contain little information on $\mathbf{G}$ as they might have high entropy.

### C.2.2  Influence of task difficulty

As the ground-truth generating factors are accessible for these vision datasets, we could explore how the difficulty of the task, i.e., $\alpha$, influence the performance gap among different methods. As shown in Table 6, SEM-IL brings a significant enhancement when $\alpha$ is not too big nor too small. When $\alpha$ is big, the test loss might be very small and there is no room to make considerable improvement. On another extreme, if $\alpha$ is too small, some attributes might be never observed during training, which is hard for the model to extract correct generating factors. Remember we expect to generalize to "red circle" by knowing the concept of red and circle from other combinations: if there is only "blue" and "box" in the training set, it is impractical for a model to learn such a concept (maybe the model can extrapolate, but that is another topic). We speculate most real tasks are in a relatively small $\alpha$ regime, as the generating factors and their possible values can be very large. Please note that as the train/test split and GenY all depend on the random seeds, the variance of the numbers in these tables could be large. However, we observe the SEM-IL consistently outperforms other methods under each task generated by different random seeds.

Table 6: Relative improvement comparison for different $\alpha$ on three datasets. We report the average and standard error of 4 different runs. $\Delta_1$ is calculated by $\frac{\text{SEMIL-Baseline}}{\text{Baseline}}$, while $\Delta_2$ is calculated by $\frac{\text{SEMIL-Baseline}}{\text{SEMIL}}$. The test MSE for different settings are the numbers multiplying $10^{-3}$. Note that the numbers of different datasets are not comparable, as the model structure and generating factors of them are different. When $\alpha$ is too small, the model fails to converge on MPI3D, which means MPI3D might be a more challenging dataset.

| | $\alpha$ | 0.8 | 0.5 | 0.2 | 0.1 | 0.02 |
|---|---|---|---|---|---|---|
| | Baseline | 3.778±0.792 | 7.902±2.000 | 28.01±11.75 | 57.87±9.852 | 355.5±136.0 |
| | NIL-only | 3.866±0.733 | 7.536±1.966 | 33.18±15.16 | 56.46±12.54 | 330.5±183.1 |
| 3dShapes | SEM-only | 2.531±0.742 | 5.15±0.415 | 21.41±5.274 | 55.48±15.76 | 292.7±148.0 |
| | SEM-IL | **0.633±0.117** | **1.27±0.112** | **5.165±0.697** | **17.52±3.103** | **221.0±122.5** |
| | Relative $\Delta_1$ | 0.8324 | **0.8396** | 0.8156 | 0.6972 | 0.3783 |
| | Relative $\Delta_2$ | 4.968 | **5.236** | 4.423 | 2.303 | 0.6086 |
| | Baseline | 45.42±10.97 | 61.95±17.80 | 125.9±29.30 | 234.0±34.94 | Not Converge |
| | NIL-only | 43.38±14.03 | 57.34±17.35 | 110.8±36.06 | 203.3±73.62 | Not Converge |
| MPI3D | SEM-only | 42.91±10.05 | 57.69±18.34 | 116.5±38.50 | 204.1±72.68 | Not Converge |
| | SEM-IL | **31.20±8.053** | **40.33±12.12** | **73.43±22.63** | **137.8±67.36** | Not Converge |
| | Relative $\Delta_1$ | 0.313 | 0.349 | **0.417** | 0.411 | - |
| | Relative $\Delta_2$ | 0.456 | 0.536 | **0.714** | 0.698 | - |
| | Baseline | 0.172±0.145 | 7.906±2.309 | 109.2±10.28 | 313.5±28.47 | 839.7±73.85 |
| | NIL-only | 0.136±0.123 | 3.678±0.869 | 56.42±7.169 | 241.0±24.68 | 630.2±32.53 |
| dSprites | SEM-only | 0.126±0.119 | 7.667±1.937 | 108.8±8.563 | 315.2±23.77 | 658.0±82.52 |
| | SEM-IL | **0.085±0.042** | **2.487±0.874** | **40.11±9.726** | **213.1±38.11** | **596.8±41.94** |
| | Relative $\Delta_1$ | 0.506 | **0.685** | 0.633 | 0.320 | 0.289 |
| | Relative $\Delta_2$ | 1.023 | **2.179** | 1.722 | 0.471 | 0.407 |

### C.3 Other baselines from disentangled representation learning

Most related works consider sys-gen as an NLP or emergent language problem rather than a general representation learning problem, so to the best of our knowledge, there are no specific advanced baselines for this concrete problem. The most related works are some VAE-based methods in disentanglement learning. In this part, we re-implement $\beta$-VAE and compare them with the baseline method in our setting. Specifically, we first pre-train the encoder of VAE on the same training set and then attach a task head for the downstream task to the "$\mu$-part" of the encoder's prediction (note that the encoder will output "$\mu$-part" and "$\sigma$-part" together). We observe that the VAE-based method performs worse than the baseline method, even though they seem to recover some disentangled factors when conducting latent traversal. This observation is consistent with the findings in [65] and [81], where the authors claim that the disentangled representations are incapable of reliably generalizing to new conceptual combinations. We also speculate that the challenging requirement of the comp-gen problem, i.e., $\mathcal{G}_{train} \cap \mathcal{G}_{test} = \emptyset$, exacerbates this: there will not be enough variations in $\mathbf{x}$ to make the VAE model capture the latent vectors precisely. However, as VAE is also an encoder-decoder system, it is possible to combine SEM-IL with it, which is left for our future work.

## D   Experimental Settings and More Results on Molecular Graph Dataset

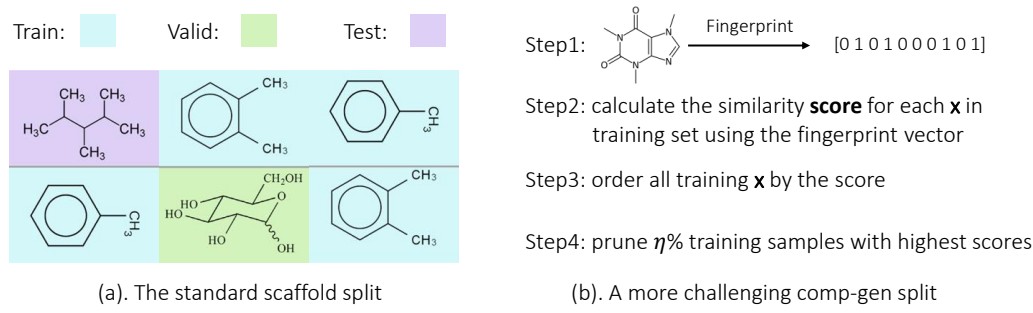

(a). The standard scaffold split                 (b). A more challenging comp-gen split

Figure 8: Left: an example of the scaffold split, the figures are copied from [79]. Right: the procedure of a more challenging few-shot split used in Table 1.

In this part, we provide an overview of the molecular graph learning dataset we used in this paper. Ogbg-molhiv and ogbg-molpcba [37] are molecular property prediction datasets proposed by MoleculeNet and then adopted by open graph benchmark (OGB) project [79]. The molhiv dataset contains roughly 40K samples, and the task is to predict whether a molecule is capable of inhibiting HIV replication (i.e., a binary classification task). The molpcba dataset is more complex, as it contains roughly 400K samples, and the target is to predict 128 different bioassays, which is a multi-task binary classification task. Ogb-PCQM4Mv2 [36] is a large-scale molecular dataset that contains roughly 4000K samples. The task is to predict the HOMO-LUMO gap (i.e., a regression task), which is a quantum physical property that is hard to calculate in traditional methods. As the test split is private, we treat the original validation split as the test split and only report the performance on it in the paper. All of the aforementioned datasets use scaffold splitting, which separates structurally different molecules into different subsets, as illustrated in the left panel in Figure 8. Under such a split, some specific structures in the test set might never occur during training, which makes it a good testbed for systematic generalization ability.

To make the task more challenging, which could simulate the scenario where $\text{supp}[P_{train}(\mathbf{G})] \cap \text{supp}[P_{test}(\mathbf{G})] = \emptyset$, we prune the training set following a procedure demonstrated in the right panel of Figure 8. Specifically, we first calculate the fingerprint of each $\mathbf{x}$ in both training, validation, and test sets using RDKit (please refer to Section 5.2 for more details). Similar to the settings used in Table 2, the fingerprint of each $\mathbf{x}$ is defined as $\text{FP}(\mathbf{x}) \in \{0, 1\}^k$. In this vector, $\text{FP}_i(\mathbf{x}) = 1$ means the molecule contain the $i$-th structure. Then, the score for each $\mathbf{x}$ in the training set is defined as how many samples in the validation and test sets share the identical $\text{FP}(\mathbf{x})$ with it. To prune the training samples which are similar to the validation and test set, we delete $\eta\%$ samples with the highest scores (for Table 1, $\eta = 50$). If we believe these 10 structures are part of $\mathbf{G}$, the remaining training samples

are more likely to have non-overlapping **G** compared with the test set, which makes the task a better testbed for systematic generalization.

The implementation of the GCN/GIN backbone used in this work is taken from the open-source code released by OGB [37]. We use the default setting of hyperparameters for all experiments (including baseline, baseline+, and interaction phase of SEM-only and SEM-IL). For the backbone structure, the depth of the GCN/GIN is 5, hidden embedding is 300, the pooling method is taking the mean, etc. For the training on downstream tasks, we use the AdamW [49] optimizer with a learning rate of $10^{-3}$, and use a cosine decay scheduler to stable the training. For the SEM layer, we search $L$ from $[10, 200]$ and $V$ from $[5, 100]$ on the validation set. For the IL-related methods, we select the imitation steps from $\{1,000; 5,000; 10,000; 50,000; 100,000\}$.

