# OpenReview forum: "Improving Compositional Generalization using Iterated Learning and Simplicial Embeddings"
_NeurIPS.cc/2023/Conference — NeurIPS 2023 poster_

### Official Review · Reviewer_7owJ · 2023-06-13

**Soundness:** 3 good
**Presentation:** 3 good
**Contribution:** 3 good
**Rating:** 5
**Confidence:** 4

**Summary:**

This paper proposes an iterated learning method with simplicial embeddings (SEM-IL) for systematic generalization.
The method is inspired by iterated learning for humans, encouraging compressibility and expressivity.
The empirical experiments show the improvement in vision tasks (known latent factors) and real molecular graph prediction tasks (unknown latent factors).
It better portrays the generation factors in vision tasks.
It also finds that the success of iterated learning may be caused by discrete messages.

**Strengths:**

- The analysis and experiments show the potential of the SEM-IL framework on systematic generalization problems.

- It is based on the cognitive science of iterated learning for humans.

**Weaknesses:**

(1) As mentioned in the conclusion section, it is still unclear why the SEM-IL method enables systematic generalization from a theoretical perspective for deep learning.

(2) More experiments will be helpful.
The paper would be more convincing with a strict theoretical explanation or comprehensive experiments.

(3) As mentioned in line 88, the decomposition into G and O is fundamentally unidentifiable because there can be different ways to define factors that work equally well on training data.
So it would be helpful to summarize what additional information SEM-IL provides to identify them.

(4) It seems source codes are not provided.

**Questions:**

Please address the points in the weakness section.

**Limitations:**

The limitations are mentioned but not in a dedicated section.
It does not have a dedicated section for social impact.

---

> ### Author Rebuttal · Authors · 2023-08-09
>
> We appreciate the reviewer for pointing out the potential and shortcomings of our work. Please also refer to the overall response part for some common concerns.
> > Q1: As mentioned in the conclusion section …
>
> Our claim that “a theoretical understanding is still missing” might be too conservative. Under some assumptions, we do have theoretical guarantees of the behavior of neural agents involved in iterated learning. What we want to strengthen here is that the theoretical analysis for a real IL-SEM system is hard, as we don’t have enough theoretical tools to describe the behavior of a non-trivial network on a multi-label problem where the pseudo labels are sampled from some dynamic and unknown distributions (i.e., imitation phase is complex). Hence we try to highlight the intuition of the method and use carefully designed experiments to verify them.
>
> For the theoretical guarantees, we can start from the analysis of Bayesian agents in Appendix B.2. If we assume the agents involved in IL update their belief of all possible $h$ in a Bayesian way, we can then theoretically guarantee that the peak of the posterior is $h^* → argmaxP_0(h | h\in H_{eff})$, where $H_{eff}$ contains all the hypotheses that satisfy the requirements of the interaction phase. The proof is not hard; we did not provide it in this version, since we think that might distract the readers from the experiments. In this toy 2*2 example, $H_{eff}$ contains all bijections. In real applications, $H_{eff}$ contains those $h$s who are capable of accomplishing downstream tasks. In short, $H_{eff}$ is the embody of expressivity pressure, while the compressibility pressure is incorporated in $P_0(h)$. We assume the prior of the mappings are negatively correlated to its coding length, as illustrated in Table 4. This is a big assumption, which motivates us to claim that “the theory for IL-SEM is still missing.” However, this assumption is likely to be true for many intelligent agents, as it is exactly what Occam’s razor claims (see also formalizations via algorithmic information theory). We then discuss where this pressure comes from in deep learning in Appendix B.3. In this part, we find for overparameterized models, the multi-generation self-distillation can bring the simplicity bias of “active bases” to the model. Although most of the theoretical analysis in this part comes from [54], we believe our application here to explain IL is also a theoretical contribution.
>
> In summary, we can guarantee the behavior of Bayesian agents in iterated learning with the assumption of $P_0(h)$. Such an assumption could have different forms in deep learning, one of which can be proved in an overparameterized setting.
> > Q2: More experiments will be helpful …
>
> Yeah, we agree that more experiments will be helpful, but we'd also like to summarize what is shown by the experiments in the current version (please also refer to the response to Reviewer 7GSE-Q2):
>
> 1. A toy setting for the Bayesian agents is in Appendix B.2&3, which is helpful for understanding how IL works in an idealized case. We can directly observe the influence of the two pressures on the belief of different mappings.
> 2. Controlled vision tasks (section 4). With the generating factors, we can observe topsim, the learning dynamics of different pairs, and the influence of sampled pseudo labels. We cannot do such an analysis if ground truth G is inaccessible, hence we use settings where it is known; we’d like to highlight, however, that MPI3d-real in particular is a “real” problem based on actual images, not synthetic ones like dSprites (indeed quite simple) or 3dShapes (also simple, but requiring some real visual processing).
> 3. Diverse molecular graph tasks (section 5). The datasets we considered have different scales and a variety of targets. With the help of the structural features extracted by the RDKit tool, we can further verify that the representations learned via IL-SEM capture the structure of generating factors well.
>
> The general response also gives some initial experimentation with GPT2, which we will add to the revised version.
> > Q3: As mentioned in line 88 …
>
> Indeed, the decomposition into G and O is fundamentally unidentifiable. However, we believe all the correct decompositions should capture the Hamming distance relationships among samples, which is determined by the ground truth generating mechanisms. For example, no matter how we decompose a specific G (e.g., the color could be {”blue”, “red”} or RGB value), the distance between “blue box” and “blue circle” should be smaller than that of “blue box” to “red circle”: the two objects in the first pair share some similarity we care about. If one decomposition cannot capture this distance relationship, it will not help with systematic generalization. Correct decompositions should be “isomorphic” (see further discussion in the response to r5b6, Q1b).
>
> For the concern about “additional information”, our answer is that there is no additional information provided. The algorithm only **amplifies the simplicity bias** to distinguish mappings from stages 3 and 4. Recalling the toy 2*2 example from Appendix B, both systematic mappings and “holistic” mappings are bijections — mutual information between G and z cannot distinguish them. One separation, however, is the simplicity of the mapping; see the discussion in lines 941-951 for an example. For the origin of this simplicity bias in deep learning, please refer to the response to Q1, Appendix B.3, and the response to reviewer eFnA (Q5).
>
> In summary, we don’t care much about the specific decomposition of G; it is not used in the algorithm. Rather, the compressibility pressure will guide our model to the most compressed mappings. Based on Occam’s razor, we believe these mappings are likely to capture the ground truth generating mechanisms and hence are able to generalize systematically.
> > Q4: The code …
>
> We have sent an anonymous link to the code to the AC, following the instructions.

---

> > ### Comment · Area_Chair_sRjF · 2023-08-17
> > **Reviewer response needed**
> >
> > Hello Reviewer,
> >
> > The authors have endeavoured to address your comments in their rebuttal. The rebuttal phase is a key part of the NeurIPS review process. I invite you to read and respond to the author's comments as soon as possible, latest tomorrow, to give everyone time to continue and conclude the discussion.
> >
> > Thank you for helping make NeurIPS a great conference for our community.

---

> > ### Comment · Reviewer_7owJ · 2023-08-21
> >
> > Thank you for the rebuttal. I raised the score.

---

### Official Review · Reviewer_7GSE · 2023-07-04

**Soundness:** 3 good
**Presentation:** 3 good
**Contribution:** 3 good
**Rating:** 5
**Confidence:** 4

**Summary:**

This work addresses the challenge deep neural networks face with systematic generalization. Systematic generalization refers to the ability to apply learned concepts to new, unobserved combinations. The authors draw on a cognitive science theory known as "iterated learning", a hypothesized process explaining how human language developed this representation ability. The theory is based on simultaneous pressures towards compressibility and expressivity when transmitting messages. The researchers propose using iterated learning, in combination with deep network models that incorporate simplicial embeddings, to generate nearly discrete messages, thereby improving systematic generalization. The effectiveness of this approach is demonstrated in vision tasks with known latent factors and in molecular graph prediction tasks with unknown latent structures, where it outperforms other methods.


**Strengths:**

1. The paper is well-written. The motivation is clear and the problems are stated clearly.

2. The problem addressed in this work is important, and could be particularly interesting to many communities such as causal learning or domain generalization.

3. The empirical performance looks promising, where the gap between the baseline and the proposed approach is non-negligible.


**Weaknesses:**

1. The method section is not described clearly. In particular, in section 3.1, the authors did not explain why IL can be used to improve the performance. Similarly, in section 3.2, I cannot understand the motivation of adopting SEM. Most of the details are deferred to Appendix B, however, it is still important to briefly explain why IL and SEM can lead to the desired results.

2. The main experiments are done with synthetic dataset such as 3dShapes or dSprites dataset. It would be interesting to see how the proposed problem can address the distribution shift problem in domain generalization benchmark such as DomainBed.

3. The model attempts to learn the minimal feature to perform prediction. This could fail when there is “short-cut” presenting in the dataset. In particular, there might be “easy feature” that is highly correlated with the true target feature, where the model could learn to use those easy features instead in the proposed setting.

4. The main theoretical results in Appendix B.3 are adopted from previous work [56], where they consider a distillation setting.

5. The main motivation is based on compressibility and expressivity. The same concept is also proposed in various previous works such as Variational Information Bottleneck (VIM), supervised contrastive learning, invariant risk minimization (IRM), or works in causal representation learning. Nevertheless, there is no discussion or comparison to previous works.


**Questions:**

1. There are three analysis provided in Appendix B to explain IL. It is a bit confusing why we need so many explanation here. In particular, it is hard to related different explanation, e.g., connecting the Bayesian analysis with the KRR analysis.

2. How do we guarantee the predictor g is a good classifier in IL? Do we also train g in the IL procedure?

3. Why can’t IL alone improve the performance? According to Appendix B, it seems that IL alone can address the proposed problem in representation learning.



**Limitations:**

There is no obvious potential negative social impact of this work.

---

> ### Author Rebuttal · Authors · 2023-08-09
>
> We appreciate the reviewer for pointing out the potential of our work. Please also refer to the overall response part.
> > Q1: The method section …
>
> Thanks for highlighting this issue; as mentioned in the general response Q1, we had trouble conveying these aspects within the space limitations, but in the final version we will emphasize the most important points of the argument. For what this will roughly look like, please see the response to reviewer eFnA (Q2 to Q4).
> > Q2: The main experiments are …
>
> First, we’d like to summarize what is shown by the experiments in the current version:
>
> 1. A toy setting for the Bayesian agents in Appendix B.2&3, which is helpful for understanding how iterated learning works in an idealized case. We can directly observe the influence of the two pressures on the belief of different mappings.
> 2. Controlled vision tasks (section 4). With the generating factors, we can observe topsim, the learning dynamics, and the influence of pseudo labels. We cannot do such an analysis if ground truth G is inaccessible; we’d like to highlight, however, that MPI3d-real in particular is a “real” problem based on actual images, not synthetic ones like dSprites or 3dShapes.
> 3. Diverse molecular graph tasks (section 5). The datasets we considered have different scales and a variety of targets. With the help of the structural features extracted by the RDKit tool, we can further verify that the representations learned via IL-SEM capture the structure of generating factors well.
>
> The general response also gives some initial experimentation with GPT2.
>
> For the distribution shift on G, we would expect some improvements, because the sys-gen is indeed a hard distribution shift problem: remember the support of P(G) of train and test sets are non-overlapping in our setting, which is a big shift.
>
> However, we are not sure whether IL-SEM brings enhancement to domain adaptation problems as in DomainBed, because of their different settings. In particular, methods run on DomainBed typically know which examples come from which domain, and try to find some kind of invariance or similar properties across domains. That is, in our framework, the domain is (a) explicitly observed and (b) belongs in the “irrelevant” O factors, not in G. Our method, however, does not do much about O; it is designed for a different setting.
>
> It may be possible to build approaches for DomainBed-type problems based on IL. For instance, we could perhaps begin with a general-purpose interaction task (as in SSL), treating all semantic features as G. We could then try to explicitly identify which factors of z correspond to the domain ID, and remove them. It also could be possible to directly incorporate domain adaptation approaches into the interaction phase. This seems beyond the scope of the present submission, however.
> > Q3: The model attempts to learn …
>
> Relying on the “short-cut” features is a long-standing problem in machine learning, which is unavoidable for any learning algorithm without extra information, including IL-SEM. In this paper, we assume there is no spurious correlation between factors in O and G. However, as in the response to Q2, most methods designed to alleviate the short-cut problem would be simple to add into the interaction phase. This version of IL-SEM would then still combine compressibility pressure from the imitation phase and ideally would gain the advantages of both frameworks.
> > Q4: The main theoretical results …
>
> The analysis of Appendix B.3 is indeed built off of [56], but we would like to highlight that in addition to bringing their theoretical analysis to this setting, we also give some theoretical insight via our definition of sys-gen, the analysis of the ladder of systematicity based on mutual information, and the Bayesian agent analysis. Please also see our discussion of the theoretical contributions in our response to Reviewer 7owJ (Q1).
> > Q5: The main motivation is based on …
>
> There might be a misunderstanding here: the term “compressibility” used in this paper is different from those used in IRM or VIM. In their setting, this word means compressing information contained in X into z, corresponding to moving from stage 1 to 3. Our use of “compressibility” measures whether a bijection can be compressed to a simpler function, i.e., stage 3→4. For causal representation learning, Figure 1 is similar to a causal diagram: we can consider all factors in G causally determine the labels Y. We will add some discussions in revision.
> > Q6: There are three analyses provided …
>
> Sorry for not clarifying that. The Bayesian analysis aims to propose a theoretical guarantee for the converging behavior of the agents involved in iterated learning. However, this theory is not perfect: one of the most important facts about compressibility pressure, the prior distribution $P_0(h)$, must be manually designed. Appendix B.3 justifies where this bias comes from when using gradient descent in deep learning without an explicit prior. The KRR in this part explains that in a simplified setting, the compressibility pressure can be considered as regularizing the “active basis” used to explain the training data, which is one form of compressibility.
> > Q7: How do we guarantee the predictor …
>
> Yes, $g$ will be trained together with $f$ in the interaction phase. We can treat the interaction phase here as a standard supervised learning process – the whole $g\circ f$ is differentiable thanks to SEM – with a special initialization of the backbone part. We don’t use $g$ in the imitation phase.
> > Q8: Why can’t IL alone improve …
>
> The main difference between IL-only and IL-SEM is the representation space. The discrete message will amplify the compressibility pressure for different reasons. First, it enables us to use the sampled pseudo labels from the teacher during imitation (discussed in section 4.2). Second, it enables us to use cross-entropy loss rather than MSE loss during imitation (discussed in Appendix B.3).

---

> > ### Comment · Reviewer_7GSE · 2023-08-17
> > **Response**
> >
> > Thanks for the rebuttal and additional results. Nevertheless, I still share similar concerns mentioned by other reviewers. For Q4-6, the definitions and the theory are not completely coherent, and one needs to consider various frameworks, e.g., Bayesian and non-Bayesian theory to understand those stages. Q8 is also not being well-addressed, as the other reviewers also stated. Discrete feature has been well-adopted to improve performance, however, combining IL with it does not means IL is the key to improving performance. Overall, I think some presentations and clarifications could be improved, and I will keep my score.

---

> > > ### Author Response · Authors · 2023-08-17
> > > **Concerns about theory and Q8.**
> > >
> > > Thanks for the reviewer's feedback. We agree that the definitions and theories are not completely coherent, which weakens the paper's contribution. While we recognize that an integrated and self-contained theoretical framework would undoubtedly bolster the paper's strength, we must admit its complexity (and many widely used methods in deep learning need such theories as well). Nevertheless, we believe considering various frameworks is not necessarily a bad thing: some readers might favor describing the same phenomenon from different perspectives, which can make the claim more persuasive. So we put them in the appendix for readers who are interested in IL.
> > >
> > > For the concerns about Q8, i.e., proving IL is the key to improving performance. We have ablation studies showing that IL+SEM outperforms IL-only, SEM-only, and baseline. When G is given, we also make a prediction on how the confidence of different message pairs evolves and verify that using experiments. We believe these observations are sufficient to claim that under the experimental setting, IL is the key to improving performance. We would appreciate it if the reviewer could give some suggestions (e.g., experimental designs) on how to prove the role of IL.
> > >
> > > By the way, the original Q8 is asking why IL alone cannot make improvements, which is different from "why IL is the key to improving the performance".

---

### Official Review · Reviewer_r5b6 · 2023-07-05

**Soundness:** 4 excellent
**Presentation:** 3 good
**Contribution:** 2 fair
**Rating:** 6
**Confidence:** 3

**Summary:**

This paper aims to develop methods to improve systematic generalization. The paper proposes several theoretical criteria related to representation learning to improve systematic generalization. Motivated by these principles, the paper then studies whether two methods, iterative learning (IL) and simplicial embeddings (SEM), and their combination can improve systematic generalization. They evaluate these methods on synthetic vision tasks and several tasks related to modeling molecules. They find that the combination of these two methods leads to the highest performance, on average, across these tasks.

**Strengths:**

* I appreciated the formal treatment of the problem in section 2, the intuition offered in section 3, and the empirical analysis connecting the proposed methods to the various hypotheses in these sections. Otherwise it would have been non-obvious how SEM and IL would be expected to improve systematic generalization.
* The paper is well written and does a nice job connecting various bodies of work. While the empirical results for the specific proposed methods are not overly convincing (see weaknesses), I could see this work potentially inspiring new ideas.
* Different from prior work on systematic generalization that proposes specialized models with task-specific inductive biases, SEM and IL are relatively general methods that do not necessarily limit the expressiveness of the underlying model.


**Weaknesses:**

* While I liked the intuition-building and "ladder of systematicity", I have two concerns with it. First, Stages 1 to 3 seem to directly follow from prior work, e.g. the information bottleneck principle (to the author's credit, this appears to be acknowledged in the appendix). Second, maybe misunderstanding on my part, but I was not fully satisfied with the definition of stage 4. As informal intuition, it makes sense that we would want to learn representations `z` that are similarly "structured" to the latent factors of the data generating procedure `G`. But it wasn't clear to me which operations are expected to be preserved through the proposed "isomorphism". Additionally, while `G` informally represents "semantic factors" it wasn't clear if this can be translated to a formal constraint on the structure of `G`, which we want `z` to be isomorphic to? I think it's fine to leave this mostly informal, but since the paper seemed to attempt to formalize this, I was left a bit unsatisfied.
* It seems like most of the intuition offered for SEM and IL corresponds to stages 1 to 3 of the ladder, which per the first point is closely aligned with criteria proposed by prior work. It wasn't clear why these methods should help achieve stage 4, which seemed more unique to the systematic generalization problem setting.
* The proposed methods seem to assume a `z` with fixed dimensionality. While this seems reasonable for one of the motivating examples (e.g. `z` representing color and shape), it is less clear how this could be applied to systematic generalization in natural language, which is referenced multiple times in the text. Prior work on systematic generalization in natural language has often considered dynamically-sized tree-structured latent variables.
* The overall effectiveness of SEM and IL is hard to gauge. The experiments show improvements on synthetic vision tasks and 3 tasks related to modeling molecules. However, it's unclear whether the proposed methods are effective on real-world vision or NLP tasks that are more popularly studied within the ML community.



**Questions:**

See weaknesses above.

**Limitations:**

I more explicit discussion of limitations could be useful.

---

> ### Author Rebuttal · Authors · 2023-08-09
>
> We appreciate the reviewer for pointing out the potential of our work. Please also refer to the overall response.
> > Q1a: While I liked …
>
> The analysis of this part originates from the information bottleneck principle (we will make this clearer in revision), but there are also some distinctions. This principle is proposed to explain different stages of GD-based supervised learning, while our ladder applies it to rank the capability of the representations learned via different methods (e.g., supervised learning, SSL, etc.) The goal of this analysis is to propose stage 4.
> > Q1b: Second, maybe misunderstanding …
>
> For stage 3→4, we believe the toy 2*2 example mentioned in Appendix A.4 and B.2 can provide a good intuition. Compare sys-mappings to arbitrary (holistic) bijections. As a sys-mapping can be decomposed into some shared rules while a holistic one cannot (see Table 4), the former can generalize to unseen combinations, while the latter cannot. Such rule **decomposition** is the **isomorphism** mentioned by the reviewer: imagining we have 2 attributes, each with 4 possible values, a sys-mapping can be decomposed into 9 rules while a holistic one needs 16 rules. Based on Occam's razor (simplicity bias), the ground truth generating mechanism is likely to have only 9 rules.
>
> Regarding the formal definition of this isomorphism, we do have one using the Wreath product in group theory (the detailed definition of Hypothesis 1 is its informal form). However, this definition requires the same $|G_i|$ for different i, and also different $G_i$ have to be independent of each other. Under this definition, we can formally prove that the isometry group of mappings is Hamming-distance preserving, which means the topological similarity between G and z of a sys-mapping should be highest among all possible groups (this can be proved using Corollary 4.2 of [Panek and Panek (2017)](https://arxiv.org/abs/1705.09987)).
>
> In the end, we deleted this proof from the submission, for the following two reasons. First, we don’t want to make strong assumptions on $G$ which aren’t satisfied by real problems, e.g. in graph problems where $G$ has complex structures and dependencies, since these assumptions are not necessary for IL: compressibility pressure still exists when $G$ is complex. Second, although we have a formal definition of the problem in the simplified setting, we find our analysis doesn’t depend on that. We thought that including these formal results in a simple setting might distract the readers from the greater generality of our intuitive analysis. This setting is similar to that of [34] for disentanglement, which we believe has similar issues. We will restore these results in the appendix in revision if it would be helpful, however.
> > Q2. It seems like …
>
> There might be a misunderstanding here. As discussed in Q1b, we know the main difference between mappings of stages 3 and 4 is how well they can be compressed. But this compression does not describe compressing information in X to representation z: it describes _compressing the mapping G→z to fewer rules_, which brings us from stage 3 to stage 4. The term “compressibility” is probably the source of this misunderstanding; we will clarify this in revision. This compressibility pressure is assumed to be “built-in” in the human cognition system, and has different origins in deep learning. Please refer to Append B.3 for more details.
>
> > Q3. The proposed method …
>
> First, the fixed $z$ won’t harm the performance too much as long as the gap between the dimensions of $z$ and $G$ is not so big: some dimensions of $z$ can encode multiple features or noise. This can be verified by the graph experiments: we don’t know the ground truth $|G|$, but the performance doesn’t change much for a wide range of $|z|$ values.
>
> What indeed matters is the structure of $G$ and $z$, which comes to the second point. For text input, which might have a more complex structure on $G$, we agree that yes, the SEM bottleneck may not be enough. The SEM structure simulates a $G$ without any hierarchical structure, simply different discrete factors. For inputs with more complexly-structured $G$, it might help to upgrade the bottleneck to a more complex (e.g., attention-based) structure, as well as measuring with more complex metric such as TRE of [Andreas (2019)](https://arxiv.org/abs/1902.07181) or HSIC [27] with more complex kernels. This adds substantial complexity, however, and since the current submission already has significant complexity and major concepts to explain we thought it better to defer this to future work.
> > Q4. The overall …
>
> IL has previously proven effective in many real applications, including multi-label image classification ([Rajeswar et al. 2022](https://arxiv.org/abs/2111.12172), machine translation [54], and VQA in [83]. However, these related works lack careful discussions on why IL works and how to apply IL to more general problems; they all rely on the inherent encoder-decoder structure. On the other hand, our submission first analyzes the key building blocks of IL. To achieve this, we chose synthetic vision datasets because knowing $G$ makes analysis much easier, and graph data because we can use RDKit to extract structural information to verify what was learned in $z$. Second, we show how to convert a general representation learning problem into an encoder-decoder style for IL, which we believe is crucial to enlarge the scope of IL. By the way, the datasets we considered, though controlled, are quite “real”: MPI3D is generated by taking pictures of a real mechanical arm, and PCQM analyzes millions of real molecules.
>
> Regarding large-scale NLP tasks, many traditional solutions already have encoder-decoder structures and discrete messages, as in [54]. Directly applying IL would be fine. For the more advanced decoder-only structure, like GPT, we are still exploring the potential of IL (we tried a small example on GPT2; please see Q3 in the general response).

---

> > ### Comment · Area_Chair_sRjF · 2023-08-17
> > **Reviewer response needed**
> >
> > Hello Reviewer,
> >
> > The authors have endeavoured to address your comments in their rebuttal. The rebuttal phase is a key part of the NeurIPS review process. I invite you to read and respond to the author's comments as soon as possible, latest tomorrow, to give everyone time to continue and conclude the discussion.
> >
> > Thank you for helping make NeurIPS a great conference for our community.

---

> > ### Comment · Reviewer_r5b6 · 2023-08-18
> >
> > Thank you for your reply. I confirm my original score.

---

### Official Review · Reviewer_eFnA · 2023-07-07

**Soundness:** 2 fair
**Presentation:** 3 good
**Contribution:** 2 fair
**Rating:** 4
**Confidence:** 4

**Summary:**

Iterated learning is hypothesized to help human language develop representation ability. The paper proposes to use iterated learning with deep network models containing simplicial embeddings to obtain approximately discrete messages. They show that this combination of changes improves systematic generalization over other approaches, demonstrating these improvements both on vision tasks with well-understood latent factors and on real molecular graph prediction tasks where the latent structure is unknown.

**Strengths:**

The paper aims to improve the systematic generalization capability of deep neural networks, which is currently an important topic in the field. The writing and figures are clear and quite easy to follow.

**Weaknesses:**

Please see in Questions.

**Questions:**

1. A more detailed explanation for iterated learning, which is currently lacking in the paper, will be appreciated.

2. The intuition of using iterated learning in this paper is merely from results in cognitive science. I have not seen convincing reasons why it is appropriate to incorporate iterated learning in deep neural networks, and what limitations of previous works can be tackled by doing so. The introduction should be rewritten to highlight these points.

3. As illustrated in Figure 3 and Table 2 (and the authors also notified), iterated learning alone may not be enough. This again raises a question: Is iterated learning really necessary for this framework?

4. I have not seen a clear connection between iterated learning and simplicial embedding. As far as I understand, the main theme of the paper is about iterated learning, and simplicial embedding empirically improves the performances further (as mentioned in Section 3.2). That is to say, the combination of iterated learning and simplicial embedding in this paper may come from trial and error rather than a thorough theoretical justification. It would be beneficial if the authors can highlight this connection in their paper.

5. The paper mentions compressibility and expressivity pressures throughout the main text. Can we quantify these pressures to show that the proposed method really helps them emerge through iterated learning?

I would happily increase my score if the authors carefully tackle my concerns.

**Limitations:**

The authors have included some discussions regarding the limitations of their work in the conclusion section.

---

> ### Author Rebuttal · Authors · 2023-08-09
>
> We appreciate the reviewer’s feedback and comments. Please also refer to the overall response for some common concerns, and here is the piece-to-piece response.
> > Q1. A more detailed explanation …
>
> Thanks. We will fix it in the revision.
> > Q2. The intuition of using …
>
> We can start from the ladder of systematicity. By analyzing the information among OGXYZ, we conclude that only focusing on the mutual information between G and z is not enough for sys-gen. In the colored-MNIST example (lines 100-111), there are many possible bijections (with maximal MI), but only sys-mappings allow for sys-gen to unseen combinations of factors. Compressibility pressure finds simpler mappings, e.g. those based on fewer “bases” as discussed in lines 941-951 in the example of Appendix B.3 or those with shorter description lengths as in Table 4. In general, this should find more systematic mappings, considering arguments based on Occam’s razor or e.g. Kolmogorov complexity. Note that the term "compressibility" here does NOT refer to compressing information from X into z, as is used to go from stage 1 to 3, but to learning rules which themselves can be compressed, i.e., from stage 3 to 4.
>
> Where does compressibility pressure come from? Here are two independent lines of work that guide us to IL. One is from a line of work stemming from cognitive science but with some existing applications in deep learning: [44] claims the role of two pressures; [66] extends this to neural agents; [83] adapts this to a seemingly different application, i.e., VQA. Another line is multi-generation self-distillation, which is also known as Born Again Networks (BAN); these can be framed as IL without an interaction phase. As discussed in [56] and Appendix B.3, BAN can impose a strong regularization on the number of “active bases”, which is proven to be hard to achieve via other explicit regularizers. BAN alone lacks the expressivity pressure given by an interaction phase, though, as discussed in Appendix B.3.
> > Q3. As illustrated in Figure 3 …
>
> We certainly do not believe that iterated learning is the only conceivable way to achieve systematic generalization. The theoretical arguments of [56] establish that the way in which IL/BAN enforces compressibility pressure (via "active bases") is difficult to achieve through standard regularizers, however, giving some credence to the idea that some form of interaction is helpful. Their analysis is in a simplified setting compared to neural network learning, though, and there is no guarantee that this particular pressure is the only useful way to achieve sys-gen. In cognitive science, [Raviv et al. (2019)](https://eprints.gla.ac.uk/281145) showed a single-generation model (with its own complexities) can achieve similar results. IL, however, is certainly vital to the approach we explore in this paper (as shown by the improvements over SEM-only in our experiments).
> > Q4. I have not seen a clear connection …
>
> To apply IL in a general representation learning system, we must divide the network into an encoder and a decoder, as discussed in section 3.2. We should also discretize the messages; discrete messages allow us to sample pseudo-labels and use cross-entropy loss during the imitation phase, which substantially enhances compressibility pressure, as discussed in Section 4.2 and Appendix B.3.
>
> Previously, [66] used an LSTM encoder and decoder and trained the whole system with REINFORCE for an emergent communication task. However, we found this method to be quite unstable: non-differentiability of the learning objective leads to huge gradient variance. Similar problems exist with other training methods such as Gumbel-Softmax or straight-through estimators. Thinking about this problem led us to SEM, previously proven to be effective on SSL methods [49]. (Interestingly, [Dessì et al. (2021)](https://arxiv.org/abs/2106.04258) drew parallels between Lewis's referential game, used in iterated learning, and SimCLR, a popular SSL method.) Furthermore, as illustrated in Figure 6 of [49], SEM can help to learn representations with higher semantic coherence (i.e., higher topsim).
> This gives three different pathways to SEM:
>
> - IL → Lewis’s game → SimCLR → SSL → SEM
> - IL → discrete message → differentiable → SEM
> - IL → compressibility → high topsim → SEM.
>
> We will emphasize these connections in revision to make it clearer that this did not simply emerge from plugging together random ideas until something worked.
> > Q5. The paper mentions …
>
> Quantifying these pressures is important. Besides observing the learning behavior, it could also help to design explicit regularizers to avoid time-consuming iterative training.
>
> For expressivity, the training loss of the downstream task in the interaction phase is a good measurement. If there is no specific target task, cognitive science applications use Lewis’s game, analogous to SimCLR (mentioned above); thus conducting SSL in this phase and using its objectives is also a reasonable choice.
>
> When the ground truth G is accessible, topsim is a reasonable measurement of compressibility pressure. For unknown G, things are more difficult, but there are various possibilities for measuring the “simplicity” based on learning theory or algorithmic information theory. For instance, see the survey of [Jiang et al. (2019)](https://arxiv.org/abs/1912.02178); many of these measures correspond at least loosely to a notion of description length as illustrated in our Table 4. The number of “active bases,” as discussed by [56], is also related to “sparsity” in the function computed by the network, but is nontrivial to measure in complex settings. Learning speed might also be a good measure; it has some support in learning theory, and the integral under the curve of NLL through learning is related to compression ratio (see e.g. [Rae (2023)](https://www.youtube.com/watch?v=dO4TPJkeaaU). Our Figures 10 and 11 also show more compressed mappings are learned faster by practical networks.

---

> > ### Comment · Area_Chair_sRjF · 2023-08-17
> > **Reviewer response needed**
> >
> > Hello Reviewer,
> >
> > The authors have endeavoured to address your comments in their rebuttal. The rebuttal phase is a key part of the NeurIPS review process. I invite you to read and respond to the author's comments as soon as possible, latest tomorrow, to give everyone time to continue and conclude the discussion.
> >
> > Thank you for helping make NeurIPS a great conference for our community.

---

> > ### Comment · Reviewer_eFnA · 2023-08-22
> >
> > I thank the authors for their responses. I have raised my score.

---

### Author Rebuttal · Authors · 2023-08-09

We appreciate the reviewers’ feedback and comments, which are quite helpful for us in improving the paper. In this overall response, we summarize some common concerns from different reviewers and provide links to the corresponding responses. Some new experimental results are also discussed in this part.
> Q1. Why do we expect iterated learning to work in deep learning, and why do we choose SEM?

In short, our solution follows 4 steps:

1. From the ladder of systematicity, we expect sys-gen to need a form of simplicity bias (compressibility pressure).
2. Iterated learning can impose compressibility pressure in some simple settings (particularly, with discrete messages).
3. To extend iterated learning to general machine learning problems, we need an encoder-decoder system and discrete message space. We also need the training to be stable.
4. Using SEM in our encoder-decoder helps create a discrete message bottleneck while keeping the whole model differentiable.

There are also more subtle connections between SEM and IL that motivated us to propose this solution; please see our response to reviewer eFnA, under Q4 (as well as Q2/Q3) for more details.

This paper uses three major concepts (sys-gen, IL, and SEM) that are not widely-known to machine learning audiences. Hence, it is not easy to give all of the necessary backgrounds in 9 pages. In the current version, we probably erred too much on the side of describing the context in detail while relegating too much explanation and intuition to the appendix, which made the paper a bit hard to follow. We will do our best in revision to emphasize the main ideas behind these choices (as discussed above) in the main body, while still explaining the necessary background in enough detail.
> Q2. What are the theoretical contributions of this paper?

The theoretical contribution of this paper starts from the formal definition of the sys-gen problem (mentioned in response to Reviewer r5b6-Q1b) and the analysis of the four stages of the systematicity ladder. By analyzing the mutual information among different variables involved in the sys-gen problem, we claim that stage 3→4 needs non-trivial systematicity bias. We then analyze the behavior of Bayesian agents in iterated learning with a manually designed systematicity bias on $P_0(h)$: the behavior of these agents can be theoretically guaranteed (mentioned in response to Reviewer 7owJ-Q1). To find whether this systematicity bias exists in deep learning, we propose Appendix B.3. The KRR analysis adopted from [56] provides a theoretical guarantee of the existence of a specific form of systematicity bias, the number of “active bases”, in a simplified setting.
> Q3. Experiments

Please also refer to the response to Reviewer 7GSE-Q2. This part explains the role played by different experiments in our previous work. We didn’t consider some common vision data like CIFAR or ImageNet, because we found it hard to fit them into the sys-gen problem: they usually contain only one multi-class label, which is hard for us to extract G. Work on disentangled representation learning also rarely considers these datasets. Compared with the vision tasks, we find molecular graph is a good test-bed for sys-gen, as discussed at the beginning of section 5.

For the NLP problems, many traditional solutions already have encoder-decoder structures and discrete messages. There is no need to insert an SEM layer to discretize messages, hence directly applying IL would be fine, as discussed in [54]. For the more advanced decoder-only structure, like GPT, we are still exploring the potential of IL; we tried a small example on GPT2, shown in the attached PDF.

---

### Decision · Program_Chairs · 2023-09-21

**Decision:**

Accept (poster)

**Comment:**

This paper combines iterated learning with simplicity embeddings to develop algorithms that improve systematic generalisation over other approaches. It demonstrates benefits of this scheme on vision tasks with clear latent factors and a molecular graph prediction task.

Strengths:

-Tackles an important problem

-The empirical results show promising improvements

-The experiments show some intuitions for why these combined approaches aid systematic generalisation

Weaknesses:

-Several reviewers were unclear on the intuition for adding the SEM mechanism. (However, while better intuition is valuable, so long as the performance improvement is robustly demonstrated, it is optional)

-More experiments would be helpful in making a robust case for performance benefits, particularly in real world settings

-The paper would benefit from more complete comparison to other approaches based on compressibility ideas